# MULTI-AGENT GUIDED POLICY OPTIMIZATION

**Yueheng Li[1], Guangming Xie[1]\*, Zongqing Lu[2]\***
[1]School of Advanced Manufacturing and Robotics, Peking University
[2]School of Computer Science, Peking University
{liyueheng,xiegming,zongqing.lu}@pku.edu.cn

## ABSTRACT

Due to practical constraints such as partial observability and limited communication, Centralized Training with Decentralized Execution (CTDE) has become the dominant paradigm in cooperative Multi-Agent Reinforcement Learning (MARL). However, existing CTDE methods often underutilize centralized training or lack theoretical guarantees. We propose Multi-Agent Guided Policy Optimization (MAGPO), a novel framework that better leverages centralized training by integrating centralized guidance with decentralized execution. MAGPO uses an autoregressive joint policy for scalable, coordinated exploration and explicitly aligns it with decentralized policies to ensure deployability under partial observability. We provide theoretical guarantees of monotonic policy improvement and empirically evaluate MAGPO on 43 tasks across 6 diverse environments. Results show that MAGPO consistently outperforms strong CTDE baselines and matches or surpasses fully centralized approaches, offering a principled and practical solution for decentralized multi-agent learning.

## 1 INTRODUCTION

Cooperative Multi-Agent Reinforcement Learning (MARL) provides a powerful framework for solving complex real-world problems such as autonomous driving (Zhou et al., 2020), traffic management (Singh et al., 2020), and robot swarm coordination (Hüttenrauch et al., 2019; Zhang et al., 2021a). However, MARL faces two fundamental challenges: the exponential growth of the joint action space with the number of agents, which hinders scalability, and the requirement for decentralized execution under partial observability, which complicates policy learning.

A widely adopted solution is Centralized Training with Decentralized Execution (CTDE) (Oliehoek et al., 2008; Kraemer & Banerjee, 2016), where agents are trained using privileged global information but execute independently based on local observations. CTDE forms the foundation of many state-of-the-art MARL algorithms and typically incorporates a centralized value function to guide decentralized policies or utility functions during training. This setup allows algorithms to benefit from global context without violating the constraints of decentralized deployment.

Parallel developments in single-agent RL have explored analogous ideas under the Partially Observable Markov Decision Process (POMDP) setting (Oliehoek & Amato, 2016). Two representative paradigms have emerged. The first is *asymmetric actor–critic* (Pinto et al., 2018), where the critic has access to full state information during training while the actor is restricted to partial observations. Conceptually, this setup can be viewed as the single-agent counterpart of CTDE: centralized information is exploited in training but not used at execution time. The second paradigm is *teacher–student learning*, where a teacher policy trained with privileged information provides direct behavioral supervision to a student policy that must operate under partial observability. Unlike asymmetric actor–critic, which transfers value information, teacher–student methods transfer action-level knowledge, potentially enabling stronger guidance.

While asymmetric designs have long been embedded in CTDE-style MARL algorithms through centralized critics, the teacher–student paradigm has only recently been extended to multi-agent settings. This has led to the Centralized Teacher with Decentralized Student (CTDS) framework (Zhao et al., 2024), which augments CTDE with an explicit centralized teacher policy. In CTDS, a

---

*Corresponding authors.

teacher outputs joint actions based on the global state, and decentralized student policies are trained to imitate these actions.

Although CTDS holds the promise of leveraging centralized coordination more effectively than value-based CTDE methods, it introduces structural challenges that are particularly pronounced in MARL. First, learning a centralized teacher over the joint action space suffers from poor scalability, as the space grows exponentially with the number of agents. Second, even if a strong teacher is obtained, decentralized students may suffer from a fundamental *imitation gap* (Weihs et al., 2021). In multi-agent settings, this gap is exacerbated by *policy asymmetry*: the centralized teacher conditions on global state and joint context, whereas decentralized students must act based solely on local observations. As a result, the space of realizable decentralized joint behaviors may not contain the teacher's strategy, leading to unavoidable performance degradation.

To overcome these limitations, we propose **Multi-Agent Guided Policy Optimization (MAGPO)**, a novel framework that bridges centralized training and decentralized execution through a principled and MARL-specific design. MAGPO addresses the scalability and *policy asymmetry* problem by constraining a centralized, autoregressive guider policy to remain closely aligned with decentralized learners throughout training. The guider policy allows agents to act sequentially conditioned on previous actions, utilizing global information and coordinated data collection (Wen et al., 2022; Mahjoub et al., 2025). The alignment ensures that the coordination strategies developed under centralized supervision remain realizable by decentralized policies, thus mitigating the imitation gap that undermines prior CTDS approaches. Unlike a direct extension of single-agent GPO (Li et al., 2025), MAGPO introduces structural mechanisms tailored to multi-agent learning, including sequential joint action modeling and decentralization-aligned updates, while preserving scalability and parallelism. We provide theoretical guarantees of monotonic policy improvement and empirically evaluate MAGPO across 43 tasks in 6 diverse environments. Results show that MAGPO consistently outperforms strong CTDE baselines and even matches or exceeds fully centralized methods, establishing it as a theoretically grounded and practically deployable solution for MARL under partial observability.

## 2 BACKGROUND

### 2.1 FORMULATION

We consider Decentralized Partially Observable Markov Decision Process (Dec-POMDP) (Oliehoek & Amato, 2016) in modeling cooperative multi-agent tasks. The Dec-POMDP is characterized by the tuple $\langle \mathcal{N}, \mathcal{S}, \mathcal{A}, r, \mathcal{P}, \mathcal{O}, \mathcal{Z}, \gamma \rangle$, where $\mathcal{N}$ is the set of agents, $\mathcal{S}$ is the set of states, $\mathcal{A}$ is the set of actions, $r$ is the reward function, $\mathcal{P}$ is the transition probability function, $\mathcal{Z}$ is the individual partial observation generated by the observation function $\mathcal{O}$, and $\gamma$ is the discount factor. At each timestep, each agent $i \in \mathcal{N}$ receives a partial observation $o_i \in \mathcal{Z}$ according to $\mathcal{O}(s; i)$ at state $s \in \mathcal{S}$. Then, each agent selects an action $a_i \in \mathcal{A}$ according to its action-observation history $\tau_i \in (\mathcal{Z} \times \mathcal{A})^*$, collectively forming a joint action denoted as $\boldsymbol{a}$. The state $s$ undergoes a transition to the next state $s'$ in accordance with $\mathcal{P}(s'|s, \boldsymbol{a})$, and agents receive a shared reward $r$. Assuming an initial state distribution $\rho \in \Delta(\mathcal{S})$, the goal is to find a decentralized policy $\boldsymbol{\pi} = \{\pi_i\}_{i=1}^n$ that maximizes the expected cumulative return:

$$V_\rho(\pi) \triangleq \mathbb{E}_{s \sim \rho}[V_{\boldsymbol{\pi}}(s)] = \mathbb{E}[\sum_{t=0}^{\infty} \gamma^t r_t | s_0 \sim \rho]. \tag{1}$$

This work follows the Centralized Training with Decentralized Execution (CTDE) paradigm (Oliehoek et al., 2008; Kraemer & Banerjee, 2016). During training, CTDE allows access to global state to stabilize learning. However, during execution, each agent operates independently, relying solely on its local action-observation history.

In centralized training, we can therefore optimize a joint policy $\boldsymbol{\mu}(\boldsymbol{a}|s)$ that coordinates all agents while enabling joint exploration. To update this joint policy, we will consider the policy mirror descent (PMD) objective (Shani et al., 2019):

$$\mu^{(k+1)} = \arg\max_{\mu} \left\{ \eta_k \langle \nabla V_\rho(\mu^{(k)}), \mu \rangle - \frac{1}{1-\gamma} D_{d_\rho(\mu^{(k)})}(\mu, \mu^{(k)}) \right\}, \tag{2}$$

where $\eta_k$ is the step size, $\rho \in \Delta(\mathcal{S})$ is an arbitrary state distribution, $d_\rho(\mu^{(k)})$ is the discounted state-visitation distribution under $\mu^{(k)}$, and $D_{d_\rho(\mu^{(k)})}$ denotes the corresponding weighted Bregman divergence. Conceptually, PMD can be viewed as a class of preconditioned policy gradient methods. Choosing the Bregman divergence to be the Euclidean distance or the Kullback–Leibler (KL) divergence recovers projected policy gradient and natural policy gradient (NPG) (Kakade, 2001), respectively. Thus, PMD provides a unifying framework that encompasses many modern policy-based RL algorithms, including TRPO (Schulman et al., 2015a) and PPO (Schulman et al., 2017). We build upon this formulation to develop our theoretical results in Section 4.1.

## 2.2 RELATED WORKS

**CTDE.** CTDE methods can be broadly categorized into value-based and policy-based approaches. Value-based methods typically employ a joint value function conditioned on the global state and joint action, alongside individual utility functions based on local observations and actions. These functions often satisfy the Individual-Global-Max (IGM) principle (Son et al., 2019), ensuring that the optimal joint policy decomposes into locally optimal policies. This line of work is known as value factorization, and includes methods such as VDN (Sunehag et al., 2017), QMIX (Rashid et al., 2020), QTRAN (Son et al., 2019), QPLEX (Wang et al., 2021a), and QATTEN (Yang et al., 2020). Policy-based methods, in contrast, typically use centralized value functions to guide decentralized policies, allowing for direct extensions of single-agent policy gradient methods to multi-agent settings. Notable examples include COMA (Foerster et al., 2018), MADDPG (Lowe et al., 2017), MAA2C (Papoudakis et al., 2020) and MAPPO (Yu et al., 2022). Additionally, hybrid methods that combine value factorization with policy-based training have been proposed, such as DOP (Wang et al., 2021b), FOP (Zhang et al., 2021b), and FACMAC (Peng et al., 2021). While CTDE has achieved strong empirical performance, most existing methods leverage global information only through the value function. We refer to these as **vanilla CTDE** methods, as they do not fully exploit the potential of centralized training.

**CTDS.** More recently, researchers have explored extending the teacher-student framework from single-agent settings to multi-agent systems, leading to the Centralized Teacher with Decentralized Students (CTDS) paradigm (Zhao et al., 2024; Chen et al., 2024; Zhou et al., 2025). In this framework, a centralized teacher policy—accessing global state and acting jointly—collects high-quality trajectories and facilitates more coordinated exploration. CTDS methods offer stronger supervision than vanilla CTDE methods. However, due to observation asymmetry (Weihs et al., 2021) and policy space mismatch, the learned decentralized policies may still suffer from suboptimal performance—issues that we explore further in the next section.

**HARL.** In contrast to vanilla CTDE and CTDS methods—many of which lack theoretical guarantees—another line of research focuses on Heterogeneous Agent Reinforcement Learning (HARL), where agents are updated sequentially during training (Zhong et al., 2023). This formulation underpins algorithms such as HATRPO and HAPPO (Kuba et al., 2022) and HASAC (Liu et al., 2025). While HARL provides better theoretical guarantees and stability, it requires agents to be heterogeneous and updated one at a time. As a result, these methods lack parallelism which is important in large-scale MARL tasks and cannot exploit parameter sharing, which has proven effective in many MARL applications (Gupta et al., 2017; Terry et al., 2020; Christianos et al., 2021a).

**CTCE.** Centralized Training with Centralized Execution (CTCE) approaches treat the multi-agent system as a single-agent problem with a combinatorially large action space. Beyond directly applying single-agent RL algorithms to MARL, a promising direction in CTCE has been to use transformers (Vaswani et al., 2017) to frame multi-agent trajectories as sequences (Chen et al., 2021). This has led to the development of powerful transformer-based methods such as Updet (Hu et al., 2021), Transfqmix (Gallici et al., 2023), and other offline methods (Meng et al., 2023; Tseng et al., 2022; Zhang et al., 2022). Two representative online methods are Multi-Agent Transformer (MAT) (Wen et al., 2022) and Sable (Mahjoub et al., 2025), which currently achieve state-of-the-art performance in cooperative MARL tasks. CTCE methods offer strong theoretical guarantees (Wen et al., 2022) and impressive empirical results. However, they fall short in practical settings that demand decentralized execution, where each agent must act based solely on its local observation and policy.

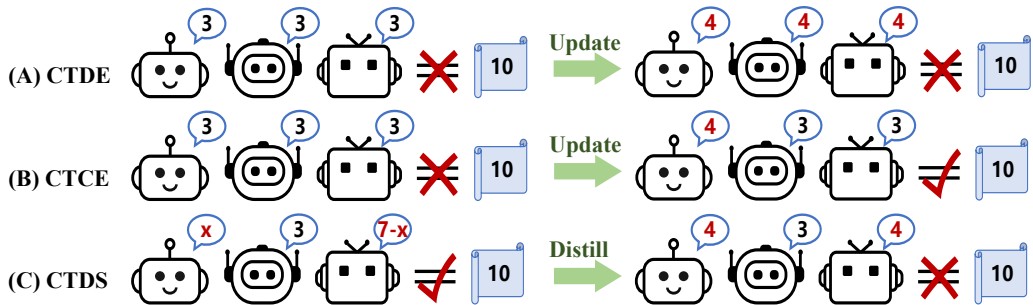

Figure 1: Illustrative example showing three different MARL settings.

## 3    PROBLEMS OF CURRENT CTDS

In this section, we examine the limitations of using a centralized teacher to supervise decentralized student policies. Two key challenges arise in this setting: **asymmetric observation spaces** and **asymmetric policy spaces**.

The first challenge—*asymmetric observation spaces*—is shared with single-agent POMDPs involving privileged information and has been extensively studied in prior work (Warrington et al., 2020; Weihs et al., 2021; Shenfeld et al., 2023; Li et al., 2025). When the teacher relies on privileged information that is unavailable (and not inferable) to the student, the student cannot faithfully reproduce the teacher's behavior. Instead, it learns to approximate the conditional expectation of the teacher's action given the observable input (Weihs et al., 2021; Warrington et al., 2020). This typically results in an "averaged" behavior that can be significantly suboptimal.

The second challenge—*asymmetric policy spaces*—is unique to the multi-agent setting. It arises from the structural mismatch between the teacher's policy (typically joint and expressive) and the students' policies (factorized and decentralized). We illustrate this challenge through a simple example shown in Figure 1. Consider a cooperative task where three agents must each output an integer such that their sum equals a target value of 10. Each agent acts once, and the system succeeds only if the total sum is exactly 10. We compare three MARL frameworks:

**(A) Vanilla CTDE**. Agents share a centralized value function but act independently via decentralized policies. Suppose all three agents use the same deterministic policy $\pi^i(\cdot|10) = 3$, producing a total of 9—which fails the task. Because each agent observes the same global state and optimizes the same objective, they may all simultaneously increase their action to 4 in the next update, producing 12, still failing. Lacking inter-agent coordination signals, the agents struggle to determine which one should adjust its action. This leads to classic miscoordination, requiring random trial-and-error exploration and memorization of rare successful configurations to eventually coordinate.

**(B) CTCE**. Agents act sequentially, observing previous agents' actions before choosing their own. Suppose the first agent updates its action to 4 and the second still picks 3. The third agent, having observed both previous actions, selects 3 and achieves the correct total. Sequential execution effectively transforms the multi-agent coordination problem into a single-agent decision-making process over a joint policy. This makes coordination straightforward and stable, without repeated random exploration. However, this setting assumes centralized execution, which is often infeasible in real-world applications requiring decentralization.

**(C) CTDS**. Now consider distilling a successful CTCE policy from (B) into decentralized student policies. If the teacher's policy is deterministic and factorizable (e.g., always producing $[4, 3, 3]$), CTDS can recover an optimal decentralized solution. However, a CTCE teacher may exploit stochastic strategies. For instance, the first agent samples $x \in \{3, 4\}$ at random, the second agent always outputs 3, and the third agent—having seen the first two actions—outputs $7 - x$, ensuring the total is always 10. While this is optimal under CTCE, it cannot be factored into independent policies: CTDS would learn two independent stochastic policies for the first and third agents, leading to failures such as $[4, 3, 4]$. If the first CTCE agent selects 3 or 4 with equal probability, the distilled decentralized policy succeeds only 50% of the time.

This example highlights the core failure mode: coordination patterns encoded in the teacher's joint policy are lost when forced into a decentralized representation. To address these limitations, we propose a new approach that constrains the teacher's policy during training, preventing it from exploiting unrepresentable coordination strategies while still allowing it to guide decentralized learners effectively.

## 4 METHOD

We introduce **Multi-Agent Guided Policy Optimization (MAGPO)**, a framework that leverages a centralized, sequentially executed guider policy to supervise decentralized learners while keeping them closely aligned. MAGPO is designed to combine the coordination benefits of centralized training with the deployment constraints of decentralized execution.

We begin by presenting the theoretical formulation and guarantee of monotonic policy improvement in the tabular setting. For simplicity, we initially assume full observability—i.e., all agents observe the global state $s$, reducing the setting to a cooperative Markov game (Littman, 1994). We will return to the partially observable case in the subsequent implementation section.

### 4.1 MULTI-AGENT GUIDED POLICY OPTIMIZATION

Our algorithm maintains a centralized guider policy with an autoregressive structure over agent actions: $\boldsymbol{\mu}(\boldsymbol{a}|s) = \mu^{i_1}(a^{i_1}|s)\mu^{i_2}(a^{i_2}|s, a^{i_1})\ldots\mu^{i_n}(a^{i_n}|s, \boldsymbol{a}^{i_{1:n-1}})$, where $i_{1:m}$ (with $m \leq n$) denotes an ordered subset $\{i_1, ..., i_m\}$ of the agent set $\mathcal{N}$, specifying the execution order. The decentralized learner policy is defined as: $\boldsymbol{\pi}(\boldsymbol{a}|s) = \prod_{j=1}^{n} \pi^{i_j}(a^{i_j}|s)$ for any ordering $i_{1:n}$, implying that all agents act independently.

Building on this structure, MAGPO optimizes the centralized guider and decentralized learner policies through an iterative four-step procedure inspired by the GPO framework (Li et al., 2025):

- **Data Collection**: Roll out the current guider policy $\boldsymbol{\mu}_k$ to collect trajectories.
- **Guider Training**: Update the guider $\boldsymbol{\mu}_k$ to $\hat{\boldsymbol{\mu}}_k$ by maximizing RL objective.
- **Learner Training**: Update the learner $\boldsymbol{\pi}_k$ to $\boldsymbol{\pi}_{k+1}$ by minimizing the KL distance $\mathrm{D}_{\mathrm{KL}}(\boldsymbol{\pi}, \hat{\boldsymbol{\mu}}_k)$.
- **Guider Backtracking**: Set $\boldsymbol{\mu}_{k+1} = \boldsymbol{\pi}_{k+1}$ for all states $s$.

The first step allows MAGPO to perform coordinated exploration using a joint policy. In the second step, the guider is updated using the Policy Mirror Descent (PMD) framework (Xiao, 2022), which solves the following optimization:

$$\hat{\boldsymbol{\mu}}_k = \arg\max_{\boldsymbol{\mu}} \left\{ \eta_k \langle Q^{\boldsymbol{\mu}_k}(s, \cdot), \boldsymbol{\mu}(\cdot|s) \rangle - \mathrm{D}_{\mathrm{KL}}(\boldsymbol{\mu}(\cdot|s), \boldsymbol{\mu}_k(\cdot|s)) \right\}, \qquad (3)$$

where $Q^{\boldsymbol{\mu}_k}$ is the Q-function of guider and $\eta_k$ is the learning rate. As discussed in Section 2.1, PMD is a general policy gradient framework that subsumes popular algorithms such as PPO and TRPO. Here, we adopt PMD for theoretical clarity and instantiate it using PPO-style updates in our practical implementation. Additional details are provided in Appendix A. In the final step, we perform guider backtracking, where the guider is reset to the current learner policy. Theoretically, this is always feasible since any decentralized policy $\boldsymbol{\pi}$ defines a valid autoregressive joint policy $\boldsymbol{\mu}$ by simply ignoring the conditioning on past actions.

Based on the framework introduced above, we have the following theorem for MAGPO.

**Theorem 4.1** (Monotonic Improvement of MAGPO). *Let $(\boldsymbol{\pi}_k)_{k=0}^{\infty}$ be the sequence of joint learner policies obtained by iteratively applying the four steps of MAGPO. Then,*

$$V_\rho(\boldsymbol{\pi}_{k+1}) \geq V_\rho(\boldsymbol{\pi}_k), \ \forall k, \qquad (4)$$

*where $V_\rho$ is the expected return under initial state distribution $\rho$.*

*Proof.* See Appendix A. □

In contrast to CTDS and standard CTDE methods like MAPPO, MAGPO provides a provable guarantee of policy improvement. This result can be understood intuitively: the guider identifies a policy that improves return in the full joint space using PMD. The learner then projects this policy into the decentralized policy space via KL minimization. Since the target was chosen via projected gradient, the resulting learner policy also improves return.

To further clarify the structure of MAGPO, we show that its learner updates can be interpreted as sequential advantage-based updates—a procedure known to ensure monotonic improvement in multi-agent settings (Kuba et al., 2022). We begin with the following lemma:

**Lemma 1** (Multi-Agent Advantage Decomposition (Kuba et al., 2022)). *In any cooperative Markov game, given a joint policy $\boldsymbol{\pi}$, for any state $s$, and any agent subset $i_{1:m}$, the following equations hold:*

$$A_{\boldsymbol{\pi}}^{i_{1:m}}(s, \boldsymbol{a}^{i_{1:m}}) = \sum_{j=1}^{m} A_{\boldsymbol{\pi}}^{i_j}(s, \boldsymbol{a}^{i_{1:j-1}}, a^{i_j}), \tag{5}$$

*where*

$$A_{\boldsymbol{\pi}}^{i_{1:m}}(s, \boldsymbol{a}^{j_{1:k}}, \boldsymbol{a}^{i_{1:m}}) \triangleq Q^{j_{1:k}, i_{1:m}}(s, , \boldsymbol{a}^{j_{1:k}}, \boldsymbol{a}^{i_{1:m}}) - Q^{j_{1:k}}(s, \boldsymbol{a}^{j_{1:k}}) \tag{6}$$

*for disjoint sets $j_{1:k}$ and $i_{1:m}$. The state-action value function for a subset is defined as*

$$Q^{i_{1:m}}(s, \boldsymbol{a}^{i_{1:m}}) \triangleq \mathbb{E}_{\boldsymbol{a}^{-i_{1:m}} \sim \boldsymbol{\pi}^{-i_{1:m}}} \left[ Q(s, \boldsymbol{a}^{i_{1:m}}, \boldsymbol{a}^{-i_{1:m}}) \right]. \tag{7}$$

Using this, we derive the following:

**Corollary 4.2** (Sequential Update of MAGPO). *The update for any individual policy $\pi^{i_j}$ with ordered subset $i_{1:j}$ can be written as:*

$$\pi_{k+1}^{i_j} = \arg\max_{\pi^{i_j}} \mathbb{E}_{\boldsymbol{a}^{i_{1:j-1}} \sim \boldsymbol{\pi}_{k+1}^{i_{1:j-1}}, a^{i_j} \sim \pi^{i_j}} \left[ A_{\boldsymbol{\pi}}^{i_j}(s, \boldsymbol{a}^{i_{1:j-1}}, a^{i_j}) \right] - \frac{1}{\eta_k} D_{KL}(\pi^{i_j}, \pi_k^{i_j}) \tag{8}$$

*Proof.* See Appendix A. $\qquad\square$

This shows that MAGPO's learner updates are equivalent to performing sequential advantage-weighted policy updates. Importantly, unlike methods such as HARL which update agents one at a time, MAGPO allows for simultaneous updates of all agent policies. This enables parallel training and improves scalability to large agent populations. Moreover, HARL requires heterogeneous agents to guarantee policy improvement, while MAGPO works with either homogeneous or heterogeneous agents, allowing it to benefit from parameter sharing—a widely adopted practice that significantly improves efficiency and generalization in MARL (Gupta et al., 2017; Terry et al., 2020; Christianos et al., 2021a).

## 4.2 Practical Implementation

In this subsection, we describe the practical implementation of MAGPO. Our implementation is based on the original GPO-clip framework (Li et al., 2025), extended to the multi-agent setting. The key difference is that the guider in MAGPO is a sequential execution policy. Since MAGPO is compatible with any autoregressive CTCE method, we do not specify the exact encoder, decoder, or attention mechanisms used. Instead, we present general training objectives for both the guider and learner components.

**Guider Update.** As introduced in the previous section, the guider policy (parameterized by $\phi$) is first optimized to maximize the RL objective, and then aligned with the learner policy. This is achieved via an RL update augmented with a KL constraint:

$$\mathcal{L}(\phi) = -\frac{1}{Tn} \sum_{j=1}^{n} \sum_{t=0}^{T-1} \left[ \min \left( r_t^{i_j}(\phi) \hat{A}_t, \text{clip}(r_t^{i_j}(\phi), \epsilon, \delta) \hat{A}_t \right) - \right.$$
$$\left. m_t^{i_j}(\delta) D_{KL} \left( \mu_\phi^{i_j}(\cdot | s_t, \boldsymbol{a}_t^{i_{1:j-1}}), \pi_\theta^{i_j}(\cdot | o_t^{i_j}) \right) \right], \tag{9}$$

where

$$r_t^{i_j}(\phi) = \frac{\mu_\phi^{i_j}(a_t^{i_j}|s_t, \boldsymbol{a}_t^{i_{1:j-1}})}{\mu_{\phi_{\text{old}}}^{i_j}(a_t^{i_j}|s_t, \boldsymbol{a}_t^{i_{1:j-1}})}, \quad m_t^{i_j}(\delta) = \mathbb{I}\left(\frac{\mu_\phi^{i_j}(a_t^{i_j}|s_t, \boldsymbol{a}_t^{i_{1:j-1}})}{\pi_\theta^{i_j}(a_t^{i_j}|o_t^{i_j})} \notin \left(\frac{1}{\delta}, \delta\right)\right),$$

and

$$\text{clip}(r_t^{i_j}(\phi), \epsilon, \delta) = \text{clip}\left(\text{clip}\left(\frac{\mu_\phi^{i_j}(a_t^{i_j}|s_t, \boldsymbol{a}_t^{i_{1:j-1}})}{\pi_\theta^{i_j}(a_t^{i_j}|o_t^{i_j})}, \frac{1}{\delta}, \delta\right) \frac{\pi_\theta^{i_j}(a_t^{i_j}|o_t^{i_j})}{\mu_{\phi_{\text{old}}}^{i_j}(a_t^{i_j}|s_t, \boldsymbol{a}_t^{i_{1:j-1}})}, 1-\epsilon, 1+\epsilon\right).$$

This objective has two modifications compared to the standard one: a **double clipping function** $\text{clip}(\cdot, \epsilon, \delta)$ and a **mask function** $m^{i_j}(\delta)$, both controlled by a new hyperparameter $\delta > 1$, which bounds the ratio between guider and learner policies within $(\frac{1}{\delta}, \delta)$. The inner clip in the double clipping function stops the gradient when the advantage signal encourages the guider to drift too far from the learner. The mask function ensures the KL loss is only applied when this ratio constraint is violated. The advantage estimate $\hat{A}_t$ is computed via generalized advantage estimation (GAE) (Schulman et al., 2015b) with value functions.

**Learner Update.** The learner policy $\boldsymbol{\pi}$, parameterized by $\theta$, is updated with two objectives: (i) behavior cloning toward the guider policy, and (ii) an RL auxiliary term to directly improve return from the collected trajectories.

$$\mathcal{L}(\theta) = \frac{1}{Tn} \sum_{j=1}^{n} \sum_{t=0}^{T-1} \left[ D_{\text{KL}}\left(\pi_\theta^{i_j}(\cdot|o_t^{i_j}), \mu_\phi^{i_j}(\cdot|s_t, \boldsymbol{a}_t^{i_{1:j-1}})\right) - \right. \tag{10}$$
$$\left. \lambda \min\left(r_t^{i_j}(\theta)\hat{A}_t, \text{clip}(r_t^{i_j}(\theta), 1-\epsilon, 1+\epsilon)\hat{A}_t\right)\right],$$

where

$$r_t^{i_j}(\theta) = \frac{\pi_\theta^{i_j}(a_t^{i_j}|o_t^{i_j})}{\mu_{\phi_{\text{old}}}^{i_j}(a_t^{i_j}|s_t, \boldsymbol{a}_t^{i_{1:j-1}})}. \tag{11}$$

The auxiliary RL objective helps maximize the utility of collected trajectories. Since the behavior policy (guider) is kept close to the learner, this term approximates an on-policy objective. In principle, we could apply sequential updates to each individual policy—analogous to HAPPO—to preserve the theoretical guarantees of monotonic improvement. However, this makes the learner updates non-parallelizable and incompatible with parameter sharing. Since no performance benefits are observed from using HAPPO, we instead adopt a MAPPO-style update: all learners share parameters and are updated jointly and in parallel. The auxiliary RL term can be treated as optional and controlled by $\lambda$.

## 5 EXPERIMENTS

We evaluate MAGPO by comparing it against several SOTA baseline algorithms from the literature. Specifically, we consider two CTCE methods—Sable (Mahjoub et al., 2025) and MAT (Wen et al., 2022)—two CTDE baselines—MAPPO (Yu et al., 2022) and HAPPO (Kuba et al., 2022)—and a vanilla implementation of on-policy CTDS, which can be viewed as MAGPO without double clipping, masking, and the RL auxiliary loss. For the joint policy in both MAGPO and CTDS, we use Sable as the default backbone. All algorithms are implemented using the JAX-based MARL library Mava (de Kock et al., 2023).

**Evaluation & Hyperparameters.** We follow the evaluation protocol from Mahjoub et al. (2025). Each algorithm is trained with 10 independent seeds per task. Training is conducted for 20 million environment steps, with 122 evenly spaced evaluation checkpoints. At each checkpoint, we record the task-specific metrics (e.g., mean episode return and win rate) over 32 evaluation episodes. For task-level results, we report the mean and 95% confidence intervals. For aggregate performance across entire environment suites, we report the min-max normalized interquartile mean (IQM) with 95% stratified bootstrap confidence intervals. The hyperparameters are tuned on each task for each algorithm, which are detailed in Appendix C.3.

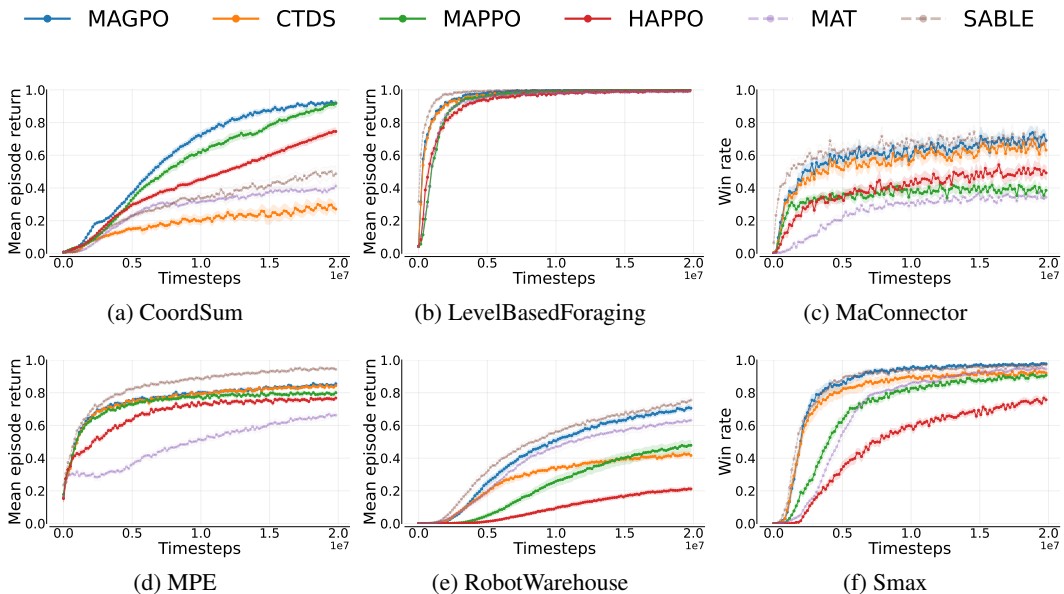

Figure 2: The sample efficiency curves aggregated per environment suite, where dashed lines represent the CTCE methods. For each environment, results are aggregated over all tasks and the min–max normalized inter-quartile mean with 95% stratified bootstrap confidence intervals are shown.

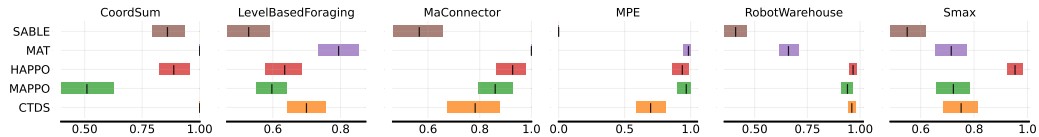

Figure 3: The overall aggregated probability of improvement for MAGPO compared to other baselines for that specific environment. A score of more than 0.5 where confidence intervals are also greater than 0.5 indicates statistically significant improvement over a baseline for a given environment (Agarwal et al., 2021).

**Environments.** We evaluate MAGPO on a diverse suite of JAX-based multi-agent benchmarks, including 4 tasks in CoordSum (introduced in this paper), 7 tasks in Level-based foraging (LBF) (Christianos et al., 2021b), 4 tasks in Connector (Bonnet et al., 2024), 3 tasks in the Multi-agent Particle Environment (MPE) (Lowe et al., 2017), 15 tasks in Robotic Warehouse (RWARE) (Papoudakis et al., 2020), and 10 tasks in The StarCraft Multi-Agent Challenge in JAX (SMAX) (Rutherford et al., 2024). The CoordSum environment, introduced in this paper, reflects the didactic examples discussed in Section 3, where agents must coordinate to output integers that sum to a given target without using fixed strategies. A detailed description is provided in Appendix B.

## 5.1 MAIN RESULTS

Figure 2 presents the per-environment aggregated sample-efficiency curves. Our results show that MAGPO achieves state-of-the-art performance across all CTDE methods and even outperforms CTCE methods on a subset of tasks. Specifically, MAGPO surpasses all CTDE baselines on 32 out of 43 tasks, and outperforms all baselines on 20 out of 43 tasks. Figure 3 reports the probability of improvement of MAGPO over other baselines. MAGPO emerges as the most competitive CTDE method and performs comparably to the SOTA CTCE method Sable in three benchmark environments. Comparing MAGPO to CTDS reveals a significant performance gap in the CoordSum and RWARE domains, suggesting that in these environments, the CTCE teacher may learn policies that are not decentralizable—rendering direct policy distillation ineffective. Additional tabular results and environment/task-level aggregation plots are provided in Appendix C.2.

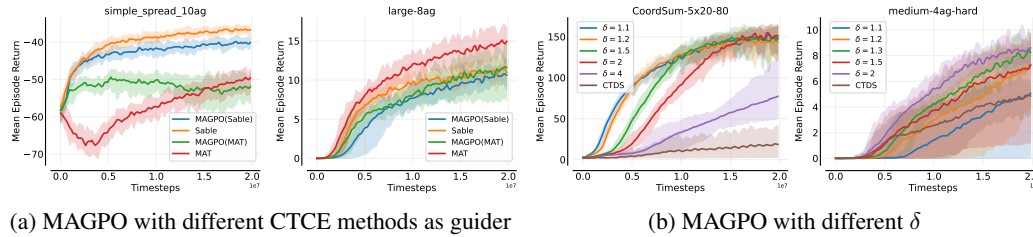

(a) MAGPO with different CTCE methods as guider  (b) MAGPO with different $\delta$

Figure 4: MAGPO performance varies with the choice of guider and the regularization ratio $\delta$.

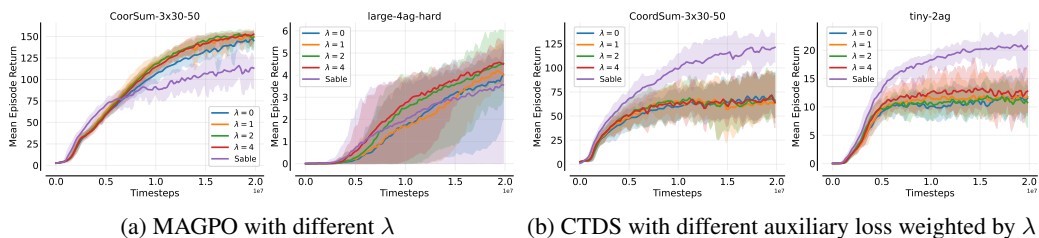

(a) MAGPO with different $\lambda$  (b) CTDS with different auxiliary loss weighted by $\lambda$

Figure 5: The effect of RL auxiliary loss.

## 5.2 ABLATIONS AND DISCUSSIONS

In this subsection, we discuss several key aspects and design choices of MAGPO.

**Bridging CTCE and CTDE.** MAGPO's performance intuitively depends heavily on the capability of the guider, which corresponds to the performance of the underlying CTCE method. In Figure 4(a), we evaluate MAGPO on two tasks where Sable and MAT exhibit different performance. In *simple_spread_10ag*, MAT performs significantly worse, resulting in poor performance of MAGPO when using MAT as the guider. In contrast, on *large-8ag*, MAT outperforms Sable, leading to better performance of MAGPO with MAT. This dependency could be seen as a limitation, but it actuall serves as a core feature: MAGPO effectively bridges CTCE and CTDE. In many practical applications that require decentralized policies, MAGPO enables advances in CTCE methods to directly benefit CTDE methods as well—facilitating the co-development of both paradigms.

**Effect of the Ratio $\delta$.** MAGPO introduces a hyperparameter $\delta$ to regulate the guider's deviation from the learner, which most strongly influences its performance. A smaller $\delta$ enforces a stricter constraint, keeping the guider closer to the learner policy; a larger $\delta$ allows the guider more freedom, potentially enabling it to explore regions of the policy space that are difficult or even unreachable under decentralized constraints. In Figure 4(b), we assess MAGPO's performance under varying $\delta$ values on two tasks. In *CoordSum-5x20-80*, a smaller $\delta$ yields better performance because the centralized guider tends to learn a policy that is not decentralizable, which must be restricted to improve imitability. Conversely, in *medium-4ag-hard*, the guider policy is more directly imitable, and restricting it too tightly hinders learning. These observations show the importance of tuning $\delta$ based on the task's structure and imitation feasibility.

**Effect of RL Auxiliary Loss.** MAGPO incorporates an RL auxiliary loss in the learner update to better utilize collected data and stabilize learning. This component is not as important as the $\delta$, but a properly tuned $\lambda$ can also improve performance, as shown in Figure 5(a). To understand this, consider the guider's RL objective is towards an undecentralizable direction and the learner pulls it backward (due to the imitation constraint), then this back-and-forth may repeatedly stall progress. By incorporating RL updates, the learner can "counter-supervise" the guider, helping it discover more decentralizable update directions. Furthermore, in Figure 5(b), we test applying the same RL auxiliary loss to a CTDS method. The results show limited benefit. This is because in CTDS, the behavioral policy is the teacher, which is not enforced to align with the student. If the teacher-student

gap is too large, the collected data is off-policy, thus on-policy RL loss on the student provides little benefit.

**Observation asymmetry.** While most of our analysis has focused on asymmetries in the policy and action spaces, observation asymmetry is equally critical. In the current framework, CTCE methods like MAT and Sable condition on the union of agents' partial observations, whereas individual policies are limited to their own local views. This mismatch creates an imitation gap, making direct imitation methods like CTDS fail, even when the underlying joint policy is decentralizable. MAGPO addresses this issue similar to the single-agent setting (Li et al., 2025), by controlling the divergence between the guider and the learner through the parameter $\delta$. In addition, privileged information—beyond the union of partial observations—is often available during centralized training (e.g., the true global state), although we do not explore it in this paper. Providing such privileged signals to the guider could further enhance its ability to supervise decentralized policies under partial observability.

**Model capacity.** In addition to policy and observation asymmetry, asymmetry can also arise from mismatched *model capacity* between training and deployment. In many practical systems, a high-capacity centralized model or teacher is used during training, while a compact decentralized policy must be deployed due to compute or latency constraints (e.g., large LLMs distilled for real-time inference, or vision-language-action models paired with lightweight controllers for high-frequency control). MAGPO naturally accommodates this setting by explicitly constraining the centralized guider to remain imitable by decentralized actors throughout training. To evaluate this property, we conduct an additional experiment on *smacv2_5_units*, where all training-time networks (e.g., critics and centralized policies) are kept at full capacity, while only the hidden dimension of the deployable decentralized actors is reduced at evaluation time. This setup simulates a realistic distillation scenario in which a large teacher must be compressed for deployment. As shown in Figure 6, MAGPO consistently outperforms CTDS across all evaluated actor capacities, and the performance gap widens as the deployed actor becomes smaller. Compared to MAPPO, both MAGPO and CTDS—being teacher–student frameworks—transfer knowledge from large models more effectively than value-based supervision alone. Importantly, MAGPO degrades more gracefully under severe capacity constraints, indicating that constraining the teacher to remain aligned with decentralized realizability improves robustness to deployment compression.

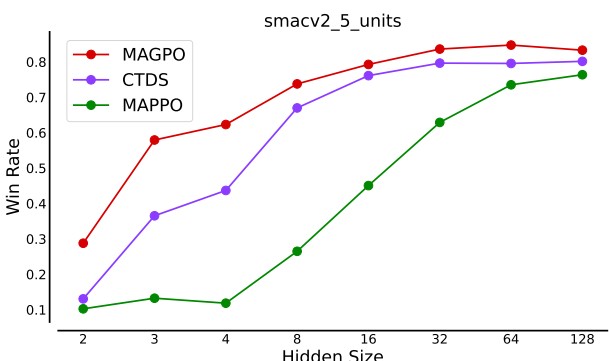

Figure 6: Effect of model capacity on performance.

## 6 CONCLUSION

We presented MAGPO, a novel framework that bridges the gap between CTCE and CTDE in cooperative MARL. MAGPO leverages a sequentially executed guider for coordinated exploration while constraining it to remain close to the decentralized learner policies. This design enables stable and effective guidance without sacrificing deployability. Our approach builds upon the principles of GPO and introduces a practical training algorithm with provable monotonic improvement. Empirical results across 43 tasks in 6 diverse environments demonstrate that MAGPO consistently outperforms state-of-the-art CTDE methods and is competitive with CTCE methods, despite relying on decentralized execution.

ACKNOWLEDGMENTS

This work was supported in part by the National Natural Science Foundation of China [grant numbers U22A2062, U23B2037, 12272008, 62450001, 62476008].

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

## THE USE OF LARGE LANGUAGE MODELS (LLMS)

LLMs are used to polish the paper writing.

## A   PROOFS

We will consider policy mirror descent (PMD) objective (Shani et al., 2019):

$$\mu^{(k+1)} = \arg\max_{\mu} \left\{ \eta_k \langle \nabla V_\rho(\mu^{(k)}), \mu \rangle - \frac{1}{1-\gamma} D_{d_\rho(\mu^{(k)})}(\mu, \mu^{(k)}) \right\}, \tag{12}$$

where $\eta_k$ is the step size, $\rho \in \Delta(\mathcal{S})$ is an arbitrary state distribution and $d_\rho(\mu^{(k)})$ is the discounted state-visitation distribution under the policy $\mu^{(k)}$, $D_{d_\rho(\mu^{(k)})}$ is the weighted Bregman divergence. Considering that

$$\langle \nabla V_\rho(\mu^{(k)}), \mu \rangle = \sum_{s \in \mathcal{S}} \langle \nabla_s V_\rho(\mu^{(k)}), \mu(\cdot|s) \rangle, \tag{13}$$

we obtain

$$\begin{aligned}
\mu^{(k+1)} &= \arg\max_{\mu} \left\{ \frac{1}{1-\gamma} \sum_{s \in \mathcal{S}} d_\rho(\mu^{(k)})(\eta_k \langle Q^{\mu^{(k)}}(s, \cdot), \mu(\cdot|s) \rangle - D(\mu(\cdot|s), \mu^{(k)}(\cdot|s))) \right\}, \\
&= \arg\max_{\mu} \left\{ \sum_{s \in \mathcal{S}} (\eta_k \langle Q^{\mu^{(k)}}(s, \cdot), \mu(\cdot|s) \rangle - D(\mu(\cdot|s), \mu^{(k)}(\cdot|s))) \right\}.
\end{aligned} \tag{14}$$

In this paper, we use KL divergence as a special case of Bregman divergence.

**Theorem A.1** (Monotonic Improvement of MAGPO). *A sequence $(\boldsymbol{\pi}_k)_{k=0}^{\infty}$ of joint policies updated by the four step of MAGPO has the monotonic property:*

$$V_\rho(\boldsymbol{\pi}_{k+1}) \geq V_\rho(\boldsymbol{\pi}_k), \ \forall k. \tag{15}$$

*Proof.* Following the derivation from Xiao (2022), the PMD objective is

$$\hat{\boldsymbol{\mu}}_k = \arg\max_{\boldsymbol{\mu}} \left\{ \eta_k \langle Q^{\boldsymbol{\mu}_k}(s, \cdot), \boldsymbol{\mu}(\cdot|s) \rangle - \mathrm{D}_{\mathrm{KL}}(\boldsymbol{\mu}(\cdot|s), \boldsymbol{\mu}_k(\cdot|s)) \right\}, \tag{16}$$

which admits the closed-form solution

$$\begin{aligned}
\hat{\boldsymbol{\mu}}_k &= \boldsymbol{\mu}_k(\boldsymbol{a}|s) \frac{\exp\left(\eta_k Q^{\boldsymbol{\mu}_k}(s, \boldsymbol{a})\right)}{z_k(s)} \\
&= \boldsymbol{\pi}_k(\boldsymbol{a}|s) \frac{\exp\left(\eta_k Q^{\boldsymbol{\pi}_k}(s, \boldsymbol{a})\right)}{z_k(s)}.
\end{aligned} \tag{17}$$

where we replace $\boldsymbol{\mu}_k$ with $\boldsymbol{\pi}_k$ due to the backtracking step.

Next, the learner update is defined as

$$\boldsymbol{\pi}_{k+1}(\cdot|s) = \arg\min_{\boldsymbol{\pi}} \mathrm{D}_{\mathrm{KL}}(\boldsymbol{\pi}(\cdot|s), \hat{\boldsymbol{\mu}}(\cdot|s)), \tag{18}$$

which guarantees the KL divergence decreases:

$$\mathrm{D}_{\mathrm{KL}}(\boldsymbol{\pi}^k(\cdot|s), \hat{\boldsymbol{\mu}}(\cdot|s)) \geq \mathrm{D}_{\mathrm{KL}}(\boldsymbol{\pi}^{k+1}(\cdot|s), \hat{\boldsymbol{\mu}}(\cdot|s)) \tag{19}$$

$$\mathbb{E}_{\boldsymbol{\pi}^k}\left[\log \boldsymbol{\pi}^k - \log \boldsymbol{\pi}^k - \eta_k Q^{\boldsymbol{\pi}^k}(s, \boldsymbol{a})\right] \geq \mathbb{E}_{\boldsymbol{\pi}^{k+1}}\left[\log \boldsymbol{\pi}^{k+1} - \log \boldsymbol{\pi}^k - \eta_k Q^{\boldsymbol{\pi}^k}(s, \boldsymbol{a})\right] \tag{20}$$

$$\eta_k \mathbb{E}_{\boldsymbol{\pi}^{k+1}}\left[Q^{\boldsymbol{\pi}^k}(s, \boldsymbol{a})\right] - \eta_k \mathbb{E}_{\boldsymbol{\pi}^k}\left[Q^{\boldsymbol{\pi}^k}(s, \boldsymbol{a})\right] \geq \mathrm{D}_{\mathrm{KL}}(\boldsymbol{\pi}^{k+1}(\cdot|s), \boldsymbol{\pi}^k(\cdot|s)) \tag{21}$$

Then, by the performance difference lemma (Kakade & Langford, 2002), we obtain:

$$\begin{aligned}
V_\rho(\boldsymbol{\pi}_{k+1}) - V_\rho(\boldsymbol{\pi}_k) &= \frac{1}{1-\gamma} \mathbb{E}_{s \sim d_\rho(\boldsymbol{\pi}_{k+1})} \left[ \mathbb{E}_{\boldsymbol{\pi}^{k+1}}\left[Q^{\boldsymbol{\pi}^k}(s, \boldsymbol{a})\right] - \mathbb{E}_{\boldsymbol{\pi}^k}\left[Q^{\boldsymbol{\pi}^k}(s, \boldsymbol{a})\right] \right] \\
&\geq \frac{1}{1-\gamma} \frac{1}{\eta_k} \mathbb{E}_{\boldsymbol{\pi}^{k+1}}\left[\mathrm{D}_{\mathrm{KL}}(\boldsymbol{\pi}^{k+1}(\cdot|s), \boldsymbol{\pi}^k(\cdot|s))\right] \\
&\geq 0.
\end{aligned} \tag{22}$$

$\square$

**Corollary A.2** (Sequential Update of MAGPO). *The update of any individual policy $\pi^{i_j}$ with any ordered subset $i_{1:j}$ can be written as:*

$$\pi_{k+1}^{i_j} = \arg\max_{\pi^{i_j}} \mathbb{E}_{a^{i_{1:j-1}} \sim \boldsymbol{\pi}_{k+1}^{i_{1:j-1}}, a^{i_j} \sim \pi^{i_j}} \left[ A_{\boldsymbol{\pi}}^{i_j}(s, \boldsymbol{a}^{i_{1:j-1}}, a^{i_j}) \right] - \frac{1}{\eta_k} D_{KL}(\pi^{i_j}, \pi_k^{i_j}) \quad (23)$$

*Proof.* We first decompose the guider policy in equation 17

$$
\begin{aligned}
\hat{\boldsymbol{\mu}}_k &= \boldsymbol{\pi}_k(\boldsymbol{a}|s) \frac{\exp\left(\eta_k Q^{\boldsymbol{\pi}_k}(s, \boldsymbol{a})\right)}{z_k(s)} \\
&= \boldsymbol{\pi}_k(\boldsymbol{a}|s) \exp\left(\eta_k Q^{\boldsymbol{\pi}_k}(s, \boldsymbol{a}) - \eta_k V^{\boldsymbol{\pi}_k}(s)\right) \frac{\exp\left(\eta_k V^{\boldsymbol{\pi}_k}(s)\right)}{z_k(s)} \\
&= \boldsymbol{\pi}_k(\boldsymbol{a}|s) \exp\left(\eta_k A^{\boldsymbol{\pi}_k}(s, \boldsymbol{a})\right) / \bar{z}_k(s) \\
&= \left(\prod_{j=1}^n \pi_k^{i_j}(a^{i_j}|s)\right) \exp\left(\eta_k \sum_{j=1}^n A_{\boldsymbol{\pi}}^{i_j}(s, \boldsymbol{a}^{i_{1:j-1}}, a^{i_j})\right) / \bar{z}_k(s) \\
&= \prod_{j=1}^n \pi_k^{i_j}(a^{i_j}|s) \frac{\exp\left(\eta_k A_{\boldsymbol{\pi}}^{i_j}(s, \boldsymbol{a}^{i_{1:j-1}}, a^{i_j})\right)}{z_k^i(s, \boldsymbol{a}^{i_{1:j-1}})}.
\end{aligned}
\quad (24)
$$

This implies that the marginal guider policy for agent $i_j$ is:

$$\hat{\mu}^{i_j}(a^{i_j}|s, \boldsymbol{a}^{i_{1:j-1}}) = \pi_k^{i_j}(a^{i_j}|s) \frac{\exp\left(\eta_k A_{\boldsymbol{\pi}}^{i_j}(s, \boldsymbol{a}^{i_{1:j-1}}, a^{i_j})\right)}{z_k^i(s, \boldsymbol{a}^{i_{1:j-1}})}. \quad (25)$$

Next, we decompose the KL divergence:

$$
\begin{aligned}
D_{KL}(\boldsymbol{\pi}(\cdot|s), \hat{\boldsymbol{\mu}}(\cdot|s)) &= \mathbb{E}_{\boldsymbol{a} \sim \boldsymbol{\pi}}\left[\log \boldsymbol{\pi}(\boldsymbol{a}|s) - \log \hat{\boldsymbol{\mu}}(\boldsymbol{a}|s)\right] \\
&= \mathbb{E}_{\boldsymbol{a} \sim \boldsymbol{\pi}}\left[\log\left(\prod_{j=1}^n \pi^{i_j}(a^{i_j}|s)\right) - \log\left(\prod_{j=1}^n \hat{\mu}^{i_j}(a^{i_j}|s, a^{i_{1:j-1}})\right)\right] \\
&= \mathbb{E}_{\boldsymbol{a} \sim \boldsymbol{\pi}}\left[\sum_{j=1}^n \log \pi^{i_j}(a^{i_j}|s) - \sum_{j=1}^n \log \hat{\mu}^{i_j}(a^{i_j}|s, a^{i_{1:j-1}})\right] \\
&= \sum_{j=1}^n \mathbb{E}_{\boldsymbol{a}^{i_{1:j-1}} \sim \boldsymbol{\pi}^{i_{1:j-1}}, a^{i_j} \sim \pi^{i_j}}\left[\log \pi^{i_j}(a^{i_j}|s) - \log \hat{\mu}^{i_j}(a^{i_j}|s, a^{i_{1:j-1}})\right].
\end{aligned}
\quad (26)
$$

Although the objective is not directly decoupled, we observe that each policy $\pi^{i_j}$ is conditionally independent of the subsequent agents given the prior ones. Therefore, we can sequentially optimize:

$$\pi_{k+1}^{i_1} = \arg\min_{\pi^{i_1}} \mathbb{E}_{a^{i_1} \sim \pi^{i_1}}\left[\log \pi^{i_1}(a^{i_1}|s) - \log \hat{\mu}^{i_1}(a^{i_1}|s)\right]$$

$$\pi_{k+1}^{i_2} = \arg\min_{\pi^{i_2}} \mathbb{E}_{a^{i_1} \sim \pi_{k+1}^{i_1}, a^{i_2} \sim \pi^{i_2}}\left[\log \pi^{i_2}(a^{i_2}|s) - \log \hat{\mu}^{i_2}(a^{i_2}|s, a^{i_1})\right]$$

$$......$$

$$\pi_{k+1}^{i_j} = \arg\min_{\pi^{i_j}} \mathbb{E}_{a^{i_{1:j-1}} \sim \boldsymbol{\pi}_{k+1}^{i_{1:j-1}}, a^{i_j} \sim \pi^{i_j}}\left[\log \pi^{i_j}(a^{i_j}|s) - \log \hat{\mu}^{i_j}(a^{i_j}|s, a^{i_{1:j-1}})\right]$$

Substituting the expression for $\hat{\mu}^{i_j}$ yields:

$$
\begin{aligned}
\pi_{k+1}^{i_j} &= \arg\min_{\pi^{i_j}} \mathbb{E}_{a^{i_{1:j-1}} \sim \boldsymbol{\pi}_{k+1}^{i_{1:j-1}}, a^{i_j} \sim \pi^{i_j}}\left[\log \pi^{i_j}(a^{i_j}|s) - \log \hat{\mu}^{i_j}(a^{i_j}|s, a^{i_{1:j-1}})\right] \\
&= \arg\min_{\pi^{i_j}} \mathbb{E}_{a^{i_{1:j-1}} \sim \boldsymbol{\pi}_{k+1}^{i_{1:j-1}}, a^{i_j} \sim \pi^{i_j}}\left[\log \pi^{i_j}(a^{i_j}|s) - \log \pi_k^{i_j}(a^{i_j}|s)) - \eta_k A_{\boldsymbol{\pi}}^{i_j}(s, \boldsymbol{a}^{i_{1:j-1}}, a^{i_j})\right] \\
&= \arg\max_{\pi^{i_j}} \mathbb{E}_{a^{i_{1:j-1}} \sim \boldsymbol{\pi}_{k+1}^{i_{1:j-1}}, a^{i_j} \sim \pi^{i_j}}\left[A_{\boldsymbol{\pi}}^{i_j}(s, \boldsymbol{a}^{i_{1:j-1}}, a^{i_j})\right] - \frac{1}{\eta_k} D_{KL}(\pi^{i_j}, \pi_k^{i_j}),
\end{aligned}
$$
$$(27)$$

which completes the proof. $\square$

# B    CoordSum Details

We introduce the **CoordSum** environment, a cooperative multi-agent benchmark designed to demonstrate the flaw of CTDS and evaluate the performance of existing paradigm. In this environment, a team of agents is tasked with selecting individual integers such that their sum matches a shared target, while avoiding repeated patterns that can be exploited by an adversarial guesser.

**Naming Convention**    Each task in the CoordSum environment is denoted as:

$$\texttt{CoordSum-<num\_agents>} \times \texttt{<num\_actions>} - \texttt{<max\_target>}$$

where `<num_agents>` is the number of agents in the team, `<num_actions>` specifies the size of each agent's discrete action space, and `<max_target>` is the maximum possible target sum.

**Observation Space**    At each timestep $t \in [1, 100]$, all agents receive the same observation: the current target value $target[t] \sim U(0, \texttt{<max\_target>})$. The observation also includes the current step count.

**Action Space**    Each agent selects an integer action from a discrete set:

$$\mathcal{A}_i = \{0, 1, \ldots, \texttt{<num\_actions>} - 1\}$$

for $i = 1, \ldots, \texttt{<num\_agents>}$. The joint action is the vector of all agents' selected integers.

**Reward Function**    To encourage agents to coordinate without relying on fixed or easily predictable strategies, the environment incorporates an opponent that attempts to guess the first agent's action using a majority vote over historical data. Specifically, for each target value, the environment records the first agent's past actions and uses the most frequent one as its guess. The reward at each timestep is defined as follows:

- If the sum of all agents' actions equals the current target:
    - A reward of **2.0** is given if the opponent's guess does *not* match the first agent's action.
    - A reward of **1.0** is given if the opponent's guess *does* match the first agent's action.
- If the sum does not match the target, a reward of **0.0** is given.

The same reward is distributed uniformly to all agents.

# C    Further experimental results

## C.1    Per-task sample efficiency results

In Figure 7, we give all task-level aggregated results. In all cases, we report the mean with 95% bootstrap confidence intervals over 10 independent runs.

## C.2    Per-task tabular results

In Table 1, we provide absolute episode metric (Colas et al., 2018; Gorsane et al., 2022) over training averaged across 10 seeds with std reported. The bolded value means the best performance across all methods, while highlighted value represents the best among CTDE methods.

## C.3    Hyperparameters

All algorithms were tuned on each task with a tuning budget of 40 trials using the Tree-structured Parzen Estimator (TPE) implemented in the Optuna library (Akiba et al., 2019). Since some of the tuned hyperparameters are provided in Mava (de Kock et al., 2023), we directly adopt them and only tune the additional algorithms and tasks. Specifically, MAGPO and HAPPO in all tasks, all algorithms in the newly introduced CoordSum tasks.

The default hyperparameters for all methods are listed in Table 2 and Table 3. The full hyperparameter search spaces are provided in Table 4, Table 5, Table 6, and Table 7.

Table 1: Performance comparison across tasks. Best overall value is bolded. Best among CTDE methods are underlined.

| | Task | MAGPO | CTDS | HAPPO | MAPPO | MAT | SABLE |
|---|---|---|---|---|---|---|---|
| CoordSum | 3x10-30 | $153.13 \pm 1.41$ | $111.98 \pm 11.15$ | $153.32 \pm 1.89$ | $\underline{\mathbf{156.37 \pm 3.56}}$ | $68.29 \pm 16.80$ | $153.70 \pm 3.77$ |
| | 3x30-50 | $156.62 \pm 1.59$ | $76.27 \pm 14.10$ | $129.35 \pm 5.22$ | $\underline{\mathbf{158.05 \pm 4.40}}$ | $87.79 \pm 8.79$ | $125.04 \pm 7.18$ |
| | 5x20-80 | $\underline{\mathbf{157.61 \pm 3.89}}$ | $19.62 \pm 11.79$ | $119.19 \pm 10.05$ | $142.51 \pm 4.87$ | $86.86 \pm 8.01$ | $48.35 \pm 15.76$ |
| | 8x15-100 | $\underline{\mathbf{129.94 \pm 8.47}}$ | $23.96 \pm 15.92$ | $77.60 \pm 5.09$ | $127.32 \pm 12.76$ | $57.22 \pm 4.97$ | $28.91 \pm 18.75$ |
| LevelBasedForaging | 15x15-3p-5f | $\underline{\mathbf{0.99 \pm 0.00}}$ | $0.96 \pm 0.02$ | $0.91 \pm 0.02$ | $0.97 \pm 0.02$ | $0.91 \pm 0.02$ | $0.96 \pm 0.01$ |
| | 15x15-4p-3f | $\underline{\mathbf{1.00 \pm 0.00}}$ | $1.00 \pm 0.00$ | $1.00 \pm 0.00$ | $1.00 \pm 0.00$ | $0.99 \pm 0.01$ | $1.00 \pm 0.00$ |
| | 15x15-4p-5f | $\underline{\mathbf{0.99 \pm 0.00}}$ | $0.99 \pm 0.00$ | $0.89 \pm 0.02$ | $0.98 \pm 0.01$ | $0.97 \pm 0.01$ | $0.99 \pm 0.00$ |
| | 2s-8x8-2p-2f-coop | $1.00 \pm 0.00$ | $1.00 \pm 0.00$ | $1.00 \pm 0.00$ | $\underline{1.00 \pm 0.00}$ | $1.00 \pm 0.00$ | $\mathbf{1.00 \pm 0.00}$ |
| | 2s-10x10-3p-3f | $0.97 \pm 0.01$ | $0.87 \pm 0.02$ | $0.99 \pm 0.01$ | $\underline{\mathbf{1.00 \pm 0.00}}$ | $0.97 \pm 0.01$ | $0.99 \pm 0.00$ |
| | 8x8-2p-2f-coop | $\mathbf{1.00 \pm 0.00}$ | $\mathbf{1.00 \pm 0.00}$ | $\mathbf{1.00 \pm 0.00}$ | $1.00 \pm 0.00$ | $1.00 \pm 0.00$ | $1.00 \pm 0.00$ |
| | 10x10-3p-3f | $1.00 \pm 0.00$ | $1.00 \pm 0.00$ | $1.00 \pm 0.00$ | $\underline{\mathbf{1.00 \pm 0.00}}$ | $0.99 \pm 0.00$ | $1.00 \pm 0.00$ |
| MaConnector | con-5x5x3a | $\underline{\mathbf{0.94 \pm 0.01}}$ | $0.93 \pm 0.02$ | $0.93 \pm 0.02$ | $0.87 \pm 0.02$ | $0.85 \pm 0.02$ | $0.92 \pm 0.02$ |
| | con-7x7x5a | $\underline{\mathbf{0.76 \pm 0.03}}$ | $0.71 \pm 0.05$ | $0.67 \pm 0.02$ | $0.63 \pm 0.02$ | $0.62 \pm 0.04$ | $0.74 \pm 0.03$ |
| | con-10x10x10a | $\underline{\mathbf{0.42 \pm 0.03}}$ | $0.37 \pm 0.04$ | $0.21 \pm 0.03$ | $0.30 \pm 0.01$ | $0.22 \pm 0.05$ | $0.39 \pm 0.03$ |
| | con-15x15x23a | $0.02 \pm 0.01$ | $0.02 \pm 0.01$ | $0.00 \pm 0.00$ | $\underline{0.02 \pm 0.01}$ | $0.00 \pm 0.00$ | $\mathbf{0.08 \pm 0.01}$ |
| MPE | spread_3ag | $\underline{-6.10 \pm 0.21}$ | $-6.34 \pm 0.29$ | $-6.63 \pm 0.49$ | $-6.72 \pm 0.22$ | $-6.59 \pm 0.23$ | $\mathbf{-4.92 \pm 0.30}$ |
| | spread_5ag | $\underline{-18.67 \pm 0.41}$ | $-20.38 \pm 0.39$ | $-23.42 \pm 0.53$ | $-22.84 \pm 0.23$ | $-25.30 \pm 1.74$ | $\mathbf{-12.75 \pm 0.91}$ |
| | spread_10ag | $-40.51 \pm 0.80$ | $\underline{-40.09 \pm 0.73}$ | $-43.68 \pm 0.55$ | $-41.83 \pm 0.52$ | $-50.07 \pm 1.72$ | $\mathbf{-36.93 \pm 0.32}$ |
| RobotWarehouse | large-4ag | $\mathbf{7.63 \pm 0.98}$ | $5.00 \pm 0.54$ | $0.61 \pm 0.54$ | $3.02 \pm 2.26$ | $4.61 \pm 0.25$ | $6.22 \pm 1.73$ |
| | large-4ag-hard | $\mathbf{4.56 \pm 0.64}$ | $2.25 \pm 1.50$ | $0.00 \pm 0.00$ | $0.00 \pm 0.01$ | $2.28 \pm 1.88$ | $3.46 \pm 1.80$ |
| | large-8ag | $\underline{10.40 \pm 0.52}$ | $7.68 \pm 0.66$ | $0.00 \pm 0.00$ | $8.35 \pm 0.66$ | $\mathbf{14.72 \pm 0.79}$ | $11.01 \pm 0.51$ |
| | large-8ag-hard | $\underline{8.66 \pm 0.61}$ | $4.32 \pm 2.86$ | $0.00 \pm 0.00$ | $3.38 \pm 3.10$ | $9.07 \pm 0.71$ | $\mathbf{9.22 \pm 0.48}$ |
| | medium-4ag | $\underline{11.46 \pm 1.16}$ | $8.22 \pm 0.56$ | $4.04 \pm 0.63$ | $7.82 \pm 3.24$ | $7.62 \pm 3.83$ | $\mathbf{12.74 \pm 1.44}$ |
| | medium-4ag-hard | $\mathbf{8.49 \pm 0.54}$ | $4.81 \pm 0.79$ | $3.75 \pm 0.63$ | $2.80 \pm 2.77$ | $4.64 \pm 2.54$ | $6.79 \pm 1.34$ |
| | medium-6ag | $\mathbf{14.78 \pm 3.28}$ | $8.49 \pm 1.07$ | $4.99 \pm 0.82$ | $12.13 \pm 0.53$ | $13.32 \pm 0.63$ | $12.97 \pm 1.03$ |
| | small-4ag | $\underline{15.09 \pm 0.71}$ | $11.07 \pm 0.81$ | $9.64 \pm 2.43$ | $10.52 \pm 0.75$ | $\mathbf{18.27 \pm 0.53}$ | $16.47 \pm 8.26$ |
| | small-4ag-hard | $\underline{11.48 \pm 0.41}$ | $7.56 \pm 1.02$ | $6.50 \pm 1.09$ | $9.44 \pm 0.35$ | $9.68 \pm 3.19$ | $\mathbf{12.02 \pm 1.20}$ |
| | tiny-2ag | $\underline{19.70 \pm 1.16}$ | $13.70 \pm 1.96$ | $10.93 \pm 2.30$ | $12.28 \pm 5.86$ | $17.06 \pm 1.61$ | $\mathbf{21.17 \pm 1.24}$ |
| | tiny-2ag-hard | $\mathbf{16.61 \pm 0.99}$ | $9.23 \pm 0.88$ | $9.61 \pm 3.09$ | $13.60 \pm 0.73$ | $13.44 \pm 2.41$ | $15.93 \pm 0.74$ |
| | tiny-4ag | $\underline{31.02 \pm 1.95}$ | $14.42 \pm 1.16$ | $17.84 \pm 2.17$ | $26.29 \pm 2.88$ | $28.19 \pm 1.02$ | $\mathbf{43.56 \pm 2.69}$ |
| | tiny-4ag-hard | $\underline{20.49 \pm 2.61}$ | $11.31 \pm 0.88$ | $16.66 \pm 2.50$ | $19.01 \pm 1.31$ | $20.54 \pm 11.99$ | $\mathbf{30.97 \pm 1.65}$ |
| | xlarge-4ag | $\mathbf{5.74 \pm 0.71}$ | $3.02 \pm 1.25$ | $0.00 \pm 0.00$ | $3.73 \pm 1.19$ | $4.71 \pm 0.43$ | $3.76 \pm 2.30$ |
| | xlarge-4ag-hard | $\underline{0.31 \pm 0.64}$ | $0.12 \pm 0.34$ | $0.00 \pm 0.00$ | $0.00 \pm 0.00$ | $0.39 \pm 1.01$ | $\mathbf{0.70 \pm 1.39}$ |
| Smax | 2s3z | $\mathbf{1.00 \pm 0.00}$ | $0.99 \pm 0.01$ | $0.99 \pm 0.01$ | $1.00 \pm 0.00$ | $0.99 \pm 0.00$ | $1.00 \pm 0.00$ |
| | 3s5z | $0.99 \pm 0.01$ | $0.98 \pm 0.01$ | $0.98 \pm 0.01$ | $\underline{1.00 \pm 0.00}$ | $\mathbf{1.00 \pm 0.00}$ | $1.00 \pm 0.01$ |
| | 3s5z_vs_3s6z | $\mathbf{0.95 \pm 0.02}$ | $0.89 \pm 0.04$ | $0.51 \pm 0.11$ | $0.91 \pm 0.05$ | $0.93 \pm 0.03$ | $0.94 \pm 0.02$ |
| | 3s_vs_5z | $0.99 \pm 0.01$ | $0.97 \pm 0.01$ | $0.95 \pm 0.01$ | $\underline{\mathbf{0.99 \pm 0.01}}$ | $0.96 \pm 0.01$ | $0.99 \pm 0.01$ |
| | 6h_vs_8z | $\mathbf{1.00 \pm 0.00}$ | $0.99 \pm 0.01$ | $0.99 \pm 0.00$ | $1.00 \pm 0.00$ | $0.99 \pm 0.01$ | $1.00 \pm 0.00$ |
| | 10m_vs_11m | $\mathbf{0.98 \pm 0.02}$ | $0.80 \pm 0.21$ | $0.14 \pm 0.03$ | $0.39 \pm 0.07$ | $0.88 \pm 0.19$ | $0.83 \pm 0.24$ |
| | 5m_vs_6m | $\underline{0.34 \pm 0.35}$ | $0.15 \pm 0.23$ | $0.03 \pm 0.01$ | $0.28 \pm 0.37$ | $\mathbf{0.68 \pm 0.36}$ | $0.59 \pm 0.41$ |
| | smacv2_5_units | $\underline{\mathbf{0.83 \pm 0.02}}$ | $0.80 \pm 0.03$ | $0.68 \pm 0.02$ | $0.76 \pm 0.02$ | $0.82 \pm 0.02$ | $0.78 \pm 0.03$ |
| | smacv2_10_units | $0.76 \pm 0.05$ | $0.65 \pm 0.06$ | $0.47 \pm 0.06$ | $\underline{\mathbf{0.76 \pm 0.02}}$ | $0.71 \pm 0.04$ | $0.65 \pm 0.04$ |
| | 27m_vs_30m | $0.99 \pm 0.01$ | $\underline{0.99 \pm 0.01}$ | $0.77 \pm 0.05$ | $0.83 \pm 0.10$ | $0.88 \pm 0.09$ | $\mathbf{1.00 \pm 0.00}$ |

Table 2: Default hyperparameters for MAT and Sable.

| Parameter | Value |
|---|---|
| Activation function | GeLU |
| Normalize Advantage | True |
| Value function coefficient | 0.1 |
| Discount | 0.99 |
| GAE | 0.9 |
| Rollout length | 128 |
| Add one-hot agent ID | True |

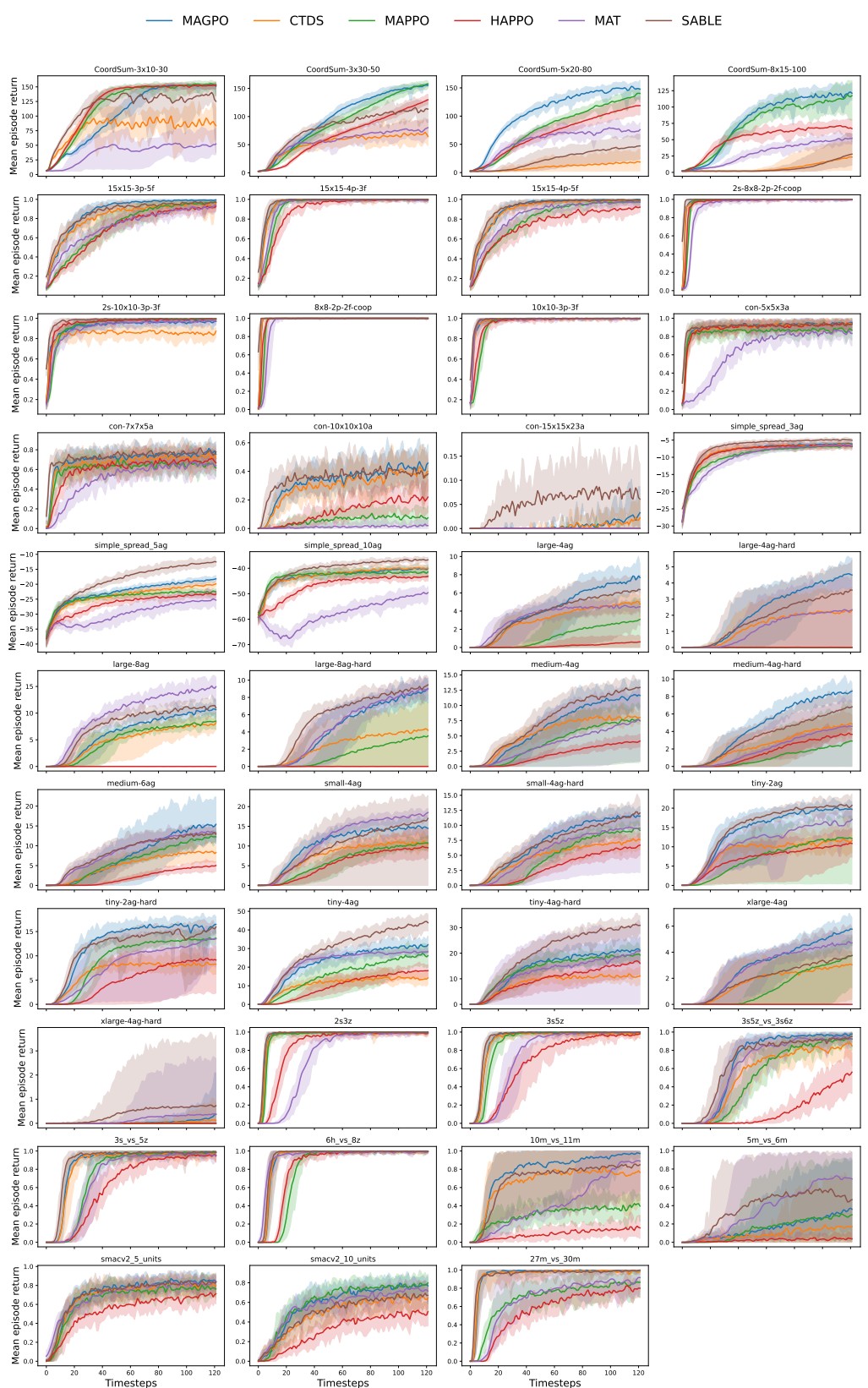

Figure 7: Mean episode return with 95% bootstrap confidence intervals on all tasks.

Table 3: Default hyperparameters for HAPPO and MAPPO.

| Parameter | Value |
|---|---|
| Value network layer sizes | [128,128] |
| Policy network layer sizes | [128,128] |
| Number of recurrent layers | 1 |
| Size of recurrent layer | 128 |
| Activation function | ReLU |
| Normalize Advantage | True |
| Value function coefficient | 0.1 |
| Discount | 0.99 |
| GAE | 0.9 |
| Rollout length | 128 |
| Add one-hot agent ID | True |

Table 4: Hyperparameter Search Space for MAT.

| Parameter | Value |
|---|---|
| PPO epochs | $\{2, 5, 10, 15\}$ |
| Number of minibatches | $\{1, 2, 4, 8\}$ |
| Entropy coefficient | $\{0.1, 0.01, 0.001, 1\}$ |
| PPO clip $\epsilon$ | $\{0.05, 0.1, 0.2\}$ |
| Maximum gradient norm | $\{0.5, 5, 10\}$ |
| Learning rate | $\{$1e-3, 5e-4, 2.5e-4, 1e-4, 1e-5$\}$ |
| Model embedding dimension | $\{32, 64, 128\}$ |
| Number of transformer heads | $\{1, 2, 4\}$ |
| Number of transformer blocks | $\{1, 2, 3\}$ |

Table 5: Hyperparameter Search Space for Sable.

| Parameter | Value |
|---|---|
| PPO epochs | $\{2, 5, 10, 15\}$ |
| Number of minibatches | $\{1, 2, 4, 8\}$ |
| Entropy coefficient | $\{0.1, 0.01, 0.001, 1\}$ |
| PPO clip $\epsilon$ | $\{0.05, 0.1, 0.2\}$ |
| Maximum gradient norm | $\{0.5, 5, 10\}$ |
| Learning rate | $\{$1e-3, 5e-4, 2.5e-4, 1e-4, 1e-5$\}$ |
| Model embedding dimension | $\{32, 64, 128\}$ |
| Number retention heads | $\{1, 2, 4\}$ |
| Number retention blocks | $\{1, 2, 3\}$ |
| Retention heads scaling parameter | $\{0.3, 0.5, 0.8, 1\}$ |

Table 6: Hyperparameter Search Space for HAPPO and MAPPO.

| Parameter | Value |
|---|---|
| PPO epochs | $\{2, 4, 8\}$ |
| Number of minibatches | $\{2, 4, 8\}$ |
| Entropy coefficient | $\{0, 0.01, 0.00001\}$ |
| PPO clip $\epsilon$ | $\{0.05, 0.1, 0.2\}$ |
| Maximum gradient norm | $\{0.5, 5, 10\}$ |
| Value Learning rate | $\{$1e-4, 2.5e-4, 5e-4$\}$ |
| Policy Learning rate | $\{$1e-4, 2.5e-4, 5e-4$\}$ |
| Recurrent chunk size | $\{8, 16, 32, 64, 128\}$ |

Table 7: Hyperparameter Search Space for MAGPO.

| Parameter | Value |
|---|---|
| Double clip $\delta$ | $\{1.1, 1.2, 1.3, 1.5, 2, 3\}$ |
| RL auxiliary loss $\lambda$ | $\{0, 1, 2, 4, 8\}$ |

