# OpenReview forum: "Multi-Agent Guided Policy Optimization"
_ICLR.cc/2026/Conference — ICLR 2026 Poster_

### Official Review · Reviewer_vuQz · 2025-10-14

**Soundness:** 3
**Presentation:** 3
**Contribution:** 2
**Rating:** 4
**Confidence:** 3

**Summary:**

The paper proposes Multi-Agent Guided Policy Optimization (MAGPO), a framework for cooperative multi-agent reinforcement learning that unites centralized training with decentralized execution. MAGPO uses a centralized autoregressive guider for coordinated exploration while keeping it aligned with decentralized policies, ensuring deployability. It offers theoretical guarantees of monotonic improvement and shows state-of-the-art results across 43 tasks, outperforming CTDE baselines and matching fully centralized methods.

**Strengths:**

The paper’s main strength lies in its conceptually coherent integration of centralized and decentralized learning in multi-agent reinforcement learning. By introducing the Multi-Agent Guided Policy Optimization (MAGPO) framework, it provides a principled way to leverage centralized guidance during training while maintaining decentralized deployability.

**Weaknesses:**

I have several concerns regarding this study, as summarized below.

1. The paper argues that the autoregressive guider policy alleviates the exponential growth of the joint action space. However, this sequential modeling still requires conditioning on all preceding agents’ actions and therefore may not scale well in the number of agents. The computational and communication costs of this approach under large agent populations are not analyzed, and all experiments are limited to small- or medium-scale settings (mostly under ten agents). Consequently, the claim of effective scalability remains largely qualitative and lacks empirical validation on truly large-scale MARL problems.

2. MAGPO’s approach to addressing policy asymmetry by regularizing the centralized guider to stay close to decentralized learners via a KL constraint (parameterized by δ). This seems to be an intuitive mechanism but not a principled solution. The theoretical guarantee of monotonic improvement holds only in the tabular and fully observable setting, leaving its applicability under partial observability and function approximation uncertain. In practice, the δ constraint appears to serve mainly as a tunable heuristic, and the paper provides limited analysis of how sensitive performance is to its choice. Thus, while MAGPO may empirically reduce the imitation gap, it does not theoretically resolve the structural mismatch between centralized and decentralized policy spaces.

3. The theoretical analysis of MAGPO assumes full observability, effectively reducing the problem to a centralized Markov game. While this simplification allows for a clean monotonic improvement proof, it undermines the practical relevance of the result. Realistic multi-agent systems operate under partial observability, where theoretical guarantees often break down due to non-Markovian local histories. Without an extension of the theory to Dec-POMDPs, the presented result is largely of limited practical utility and does not substantiate the paper’s claims of theoretical robustness under decentralized execution.

4. The role of the “guider backtracking” step (resetting the guider to the learner policy each iteration) is not well justified empirically. While it appears necessary for the theoretical monotonic improvement proof, this requirement arises only under restrictive assumptions (tabular, fully observable setting). In practice, backtracking may slow learning by erasing the guider’s accumulated coordination knowledge. The paper would benefit from an in-depth ablation study or clarification on whether strict backtracking is essential for stability, or if a softer synchronization mechanism could yield faster convergence (and better performance) without violating practical stability.

5. The addition of the RL auxiliary loss in the learner update is a pragmatic design choice rather than a theoretically required one. It helps the learner recover from the imitation gap when the guider’s centralized behavior is not perfectly realizable under decentralized observations. However, the motivation is somewhat heuristic. The paper includes a limited ablation (Fig. 5a) demonstrating that λ influences performance, but offers no systematic sensitivity or interaction analysis. In particular, the relationship between λ and the guider–learner KL constraint (δ) remains unexplored. The effect of λ is treated empirically rather than analytically, leaving uncertainty about the stability and generality of this design choice.

6. The experimental results are broad and methodologically sound, showing that MAGPO is a competitive CTDE framework. However, the improvements over strong baselines such as MAPPO, HAPPO, and Sable are often moderate rather than decisive. While MAGPO demonstrates robustness and conceptual coherence, the evidence does not clearly establish a substantial empirical leap over other SOTA methods. Particularly, MAGPO’s performance sees to heavily depend on the guider’s backbone (Sable vs. MAT) and the δ hyperparameter. This makes the results appear fragile, strong when tuned, weaker otherwise. This paper doesn’t present qualitative examples of improved coordination, communication efficiency, or emergent behaviors. So I don’t really see how MAGPO achieves its advantage.

**Questions:**

How well does the proposed autoregressive guider policy actually scale to large multi-agent systems, given that it still requires sequential conditioning over agents’ actions and has not been empirically validated beyond small- or medium-scale settings?

Since the monotonic improvement proof assumes full observability and tabular settings, how meaningful are the theoretical guarantees for realistic partially observable environments, and can the authors extend or justify the theory’s applicability to Dec-POMDPs and function-approximation cases?

To what extent are the key algorithmic components, including guider backtracking, the δ constraint, and the RL auxiliary loss (λ), essential for achieving stable learning, and how sensitive is MAGPO’s performance to these design choices and to the selection of the guider backbone (e.g., Sable vs. MAT)?

---

> ### Author Response · Authors · 2025-11-20
>
> We thank the reviewers for their constructive feedback and thoughtful suggestions. We appreciate the time and effort dedicated to evaluating our work. The comments have helped us clarify our contributions and improve the presentation. Below, we address each concern in detail.
>
> >[**Weakness 1 & Q1**] The paper argues that the autoregressive guider policy alleviates the exponential growth of the joint action space. However, this sequential modeling still requires conditioning on all preceding agents’ actions and therefore may not scale well in the number of agents. The computational and communication costs of this approach under large agent populations are not analyzed, and all experiments are limited to small- or medium-scale settings (mostly under ten agents). Consequently, the claim of effective scalability remains largely qualitative and lacks empirical validation on truly large-scale MARL problems. How well does the proposed autoregressive guider policy actually scale to large multi-agent systems, given that it still requires sequential conditioning over agents’ actions and has not been empirically validated beyond small- or medium-scale settings?
>
> We would like to clarify that our paper **does not claim** that the autoregressive guider policy _fundamentally solves_ the exponential growth of the joint action space. Rather, consistent with recent empirical findings, we argue that **autoregressive modeling is a more effective way** of handling this challenge compared with fully factorized or naive centralized approaches.
>
> Theoretically, **any centralized method must interact with the joint action space**—otherwise it becomes an independent learning method. Existing approaches differ only in how they manage joint action dependencies, typically through function approximation.
> Our method is no exception: it provides an effective way to exploit centralized training via a guider, but the exact scalability depends on the specific guider architecture adopted, paralleling the design flexibility in the MAGPO framework.
>
> For instance, the current instantiation of MAGPO uses a RetNet-style guider[1], which has $O(1)$ **inference cost** and $O(N)$ **memory complexity** for sequence length (agent number) N. This is substantially more scalable than standard Transformers, which require $O(N)$ inference cost and $O(N^2)$ memory. Therefore, the guider architecture itself can be made highly scalable in practice.
>
> Since the scalability of a particular guider design is **not the central contribution** of this work, we did not include extremely large-scale experiments. However, such scalability has already been empirically demonstrated in previous work such as Sable [2], which scales autoregressive RetNet-based backbone to over $10^3$ agents in MARL settings.
>
> [1] Retentive Network: A Successor to Transformer for Large Language Models
>
> [2] Sable: a Performant, Efficient and Scalable Sequence Model for MARL

---

> > ### Author Response · Authors · 2025-11-20
> >
> > >[**Weakness 2**] MAGPO’s approach to addressing policy asymmetry by regularizing the centralized guider to stay close to decentralized learners via a KL constraint (parameterized by δ). This seems to be an intuitive mechanism but not a principled solution. The theoretical guarantee of monotonic improvement holds only in the tabular and fully observable setting, leaving its applicability under partial observability and function approximation uncertain. In practice, the δ constraint appears to serve mainly as a tunable heuristic, and the paper provides limited analysis of how sensitive performance is to its choice. Thus, while MAGPO may empirically reduce the imitation gap, it does not theoretically resolve the structural mismatch between centralized and decentralized policy spaces.
> >
> > While we introduce our method intuitively, the proposed mechanism is in fact **principled**. The KL-constrained guider update corresponds to a **projected policy gradient / Natural Policy Gradient (NPG)** step. As shown in Theorem A.1 in the appendix, the guider is first updated via NPG and then **projected back into the decentralized learner’s policy space**. This projected-gradient structure is a standard and principled approach for solving constrained optimization problems, where here the constraint enforces that the centralized guider remains _decentralizable_—precisely the condition required to avoid the imitation gap.
> >
> > Regarding partial observability, the underlying GPO framework is **explicitly designed to handle POMDPs**. Its theoretical guarantees under partial observability follow directly from Proposition 1 of [3], which our work inherits. We discuss this in line 457 of the paper. We did not expand further because the issue already exists in the **single-agent** setting, while the unique challenge in Dec-POMDPs is policy-space asymmetry, which is the main focus of this work.
> >
> > Concerning the hyperparameter:
> >
> > - Theoretically, eliminating the imitation gap corresponds to setting $\delta=1$, i.e., taking the exact guider backtracking step.
> > - In practice, a larger $\delta$ allows the learner to better exploit the guider’s guidance.
> > - Sensitivity to hyperparameters is not unique to MAGPO; all algorithms exhibit such behavior. To ensure fairness, **all methods receive the same hyperparameter tuning budget**. An algorithm that is overly sensitive would perform inconsistently across tasks.
> > - Empirically, MAGPO is robust across all tasks, indicating that it is not excessively sensitive to $\delta$.
> >
> > [3] Guided Policy Optimization under Partial Observability.
> >
> >
> >
> > >[**Weakness 3 & Q2**] The theoretical analysis of MAGPO assumes full observability, effectively reducing the problem to a centralized Markov game. While this simplification allows for a clean monotonic improvement proof, it undermines the practical relevance of the result. Realistic multi-agent systems operate under partial observability, where theoretical guarantees often break down due to non-Markovian local histories. Without an extension of the theory to Dec-POMDPs, the presented result is largely of limited practical utility and does not substantiate the paper’s claims of theoretical robustness under decentralized execution.
> > Since the monotonic improvement proof assumes full observability and tabular settings, how meaningful are the theoretical guarantees for realistic partially observable environments, and can the authors extend or justify the theory’s applicability to Dec-POMDPs and function-approximation cases?
> >
> > First, as discussed earlier, **Theorem 4.1 directly extends to the partially observable setting**. The same argument used in the original GPO framework naturally applies here, and thus our monotonic improvement guarantee is not restricted to fully observable environments.
> >
> > Second, although Theorem 4.1 is presented in the tabular setting to clearly illustrate the core mechanism, **Corollary 4.2 provides a more practical guarantee**, analogous to the theoretical setting used in HAPPO. At the same time, MAGPO achieves **stronger properties regarding parallelization and parameter sharing**. To the best of our knowledge, no existing CTDE-based MARL algorithm provides stronger theoretical guarantees under comparably general conditions.
> >
> > Extending these results to pure Dec-POMDPs with function approximation would require additional structural assumptions such as decodability or deterministic filters [4], which are standard prerequisites in theoretical RL works studying partially observable settings. Such results would naturally fall under the scope of a dedicated RL-theory publication rather than a method-focused MARL paper. Our work aims to provide a practical and theoretically grounded framework under CTDE, consistent with existing literature.
> >
> > [4]  Provable partially observable reinforcement learningwithprivileged information.

---

> > > ### Author Response · Authors · 2025-11-20
> > >
> > > >[**Weakness 4**] The role of the “guider backtracking” step (resetting the guider to the learner policy each iteration) is not well justified empirically. While it appears necessary for the theoretical monotonic improvement proof, this requirement arises only under restrictive assumptions (tabular, fully observable setting). In practice, backtracking may slow learning by erasing the guider’s accumulated coordination knowledge. The paper would benefit from an in-depth ablation study or clarification on whether strict backtracking is essential for stability, or if a softer synchronization mechanism could yield faster convergence (and better performance) without violating practical stability.
> > >
> > > As explained above, the “guider backtracking” step is in fact a **projection step** that maps the guider back into the **decentralizable policy space**, preventing it from drifting into regions that cannot be realized by decentralized learners. While this step may slow the guider’s accumulation of coordination knowledge, it is crucial for **avoiding excessive reliance on privileged information** and, consequently, for preventing the **imitation gap**.
> > >
> > > This results in an inherent trade-off. We use the hyperparameter $\delta$ to control the strength of backtracking:
> > >
> > > - Larger $\delta$ → weaker projection → guider can exploit more centralized coordination.
> > > - $\delta\approx 1$ → strong projection → stricter control on decentralizability.
> > >
> > > Figure 4(b) illustrates this trade-off clearly. In **CoordSum**, a task specifically designed to expose imitation-gap failures, a strong projection ($\delta=1.1$) is necessary; weaker or absent projection (e.g., CTDS) performs poorly. In contrast, in tasks such as **RWARE medium-4ag-hard**, an intermediate $\delta$ achieves the best balance, though again **no-backtracking variants consistently underperform**.
> > >
> > > Thus, the backtracking step is not merely an artifact of the theory—it is a practical mechanism for maintaining decentralizability and avoiding structural failures. The empirical results demonstrate that some degree of projection is essential, while the hyperparameter $\delta$ allows flexible and effective control over its strength.
> > >
> > > >[**Weakness 5**] The addition of the RL auxiliary loss in the learner update is a pragmatic design choice rather than a theoretically required one. It helps the learner recover from the imitation gap when the guider’s centralized behavior is not perfectly realizable under decentralized observations. However, the motivation is somewhat heuristic. The paper includes a limited ablation (Fig. 5a) demonstrating that λ influences performance, but offers no systematic sensitivity or interaction analysis. In particular, the relationship between λ and the guider–learner KL constraint (δ) remains unexplored. The effect of λ is treated empirically rather than analytically, leaving uncertainty about the stability and generality of this design choice.
> > >
> > > We agree that the RL auxiliary loss is a pragmatic addition, but it is supported by two theoretical motivations:
> > > 1. **Near on-policy condition:**
> > >    Because the guider and learner remain close due to the projection step, the guider’s behavior policy is nearly on-policy for the learner. This makes the RL loss a valid and stable objective for improving the learner directly.
> > > 2. **Mitigating compounding errors in imitation learning:**
> > >    Pure KL minimization can suffer from compounding errors—two policies may have small KL divergence yet exhibit large performance gaps. The RL loss directly optimizes the learner’s return and therefore reduces dependence on the guider’s approximation error, improving stability.
> > >
> > > As Fig. 5a shows, the RL auxiliary loss **does not significantly change performance**, functioning primarily as a stabilizer rather than a critical driver of learning. While we do not provide an analytical rule for choosing $\lambda$, our experiments apply a **fixed tuning budget across all methods**, including all MAGPO hyperparameters ($\lambda$ and $\delta$). Despite having additional hyperparameters, **MAGPO remains robust across all tasks**, demonstrating that its performance is not overly sensitive or brittle.

---

> > > > ### Author Response · Authors · 2025-11-20
> > > >
> > > > >[**Weakness 6**] The experimental results are broad and methodologically sound, showing that MAGPO is a competitive CTDE framework. However, the improvements over strong baselines such as MAPPO, HAPPO, and Sable are often moderate rather than decisive. While MAGPO demonstrates robustness and conceptual coherence, the evidence does not clearly establish a substantial empirical leap over other SOTA methods. Particularly, MAGPO’s performance sees to heavily depend on the guider’s backbone (Sable vs. MAT) and the δ hyperparameter. This makes the results appear fragile, strong when tuned, weaker otherwise. This paper doesn’t present qualitative examples of improved coordination, communication efficiency, or emergent behaviors. So I don’t really see how MAGPO achieves its advantage.
> > > >
> > > > Compared with MAPPO and HAPPO, **MAGPO’s performance improvements are decisive rather than moderate**.
> > > > As shown in Figure 3, MAGPO achieves **statistically significant** gains over both MAPPO and HAPPO across _all_ benchmark tasks except MAPPO on CoordSum. In RWARE (Figure 2), MAGPO consistently outperforms MAPPO and HAPPO by a **large margin**, demonstrating clear empirical superiority in challenging cooperative domains.
> > > >
> > > > Regarding Sable, it is important to note that **Sable is a fully centralized method** and therefore enjoys a more favorable theoretical upper bound than any decentralized CTDE method. For this reason, we make no claim of universally outperforming Sable. Instead, our goal is to provide a **decentralizable and theoretically grounded CTDE framework** whose scalability can benefit from advances in centralized sequence models.
> > > >
> > > > On the concern that performance depends on the guider backbone:
> > > > this is not a weakness, but a desirable property. Modern autoregressive sequence models—such as Transformer variants and RetNet—are rapidly advancing and becoming increasingly effective for large-scale decision-making. Unlike methods that require specialized architectures under CTDE, MAGPO can directly leverage progress in centralized modeling without redesigning the decentralized learner. This creates a flexible and future-proof framework in which improvements in centralized policies immediately translate into better learner performance.
> > > >
> > > > >[**Q3**] To what extent are the key algorithmic components, including guider backtracking, the δ constraint, and the RL auxiliary loss (λ), essential for achieving stable learning, and how sensitive is MAGPO’s performance to these design choices and to the selection of the guider backbone (e.g., Sable vs. MAT)?
> > > >
> > > > Regarding the importance and sensitivity of MAGPO’s components:
> > > > 1. **Guider backbone (most essential)**
> > > >     The guider backbone is the primary driver of overall performance. This is expected—if the guider is weak, it provides poor training signals and the entire system performs suboptimally. MAGPO’s flexibility allows us to directly adopt strong, rapidly improving centralized models.
> > > >
> > > > 2. **Guider backtracking / $\delta$ constraint (second most essential)**
> > > >     This component manages the trade-off between exploiting centralized coordination and preventing imitation gap. A properly chosen $\delta$ ensures that the guider remains decentralizable while still providing useful coordination knowledge. As shown in Figure 4(b), different tasks have different optimal trade-offs, but completely removing backtracking typically harms performance.
> > > >
> > > > 3. **RL auxiliary loss ($\lambda$) (secondary)**
> > > >     The RL auxiliary objective provides mild stabilization and improves sample efficiency, but it is not the dominant factor. As shown in Fig. 5(a), MAGPO’s performance is robust across a broad range of $\lambda$, indicating that the algorithm is not overly sensitive to this hyperparameter.

---

> > > > > ### Comment · Reviewer_vuQz · 2025-11-22
> > > > >
> > > > > Thank you for the detailed responses. While I appreciate the clarifications and additional context, several of my core concerns remain insufficiently addressed.
> > > > >
> > > > > First, the scalability argument still lacks direct empirical or analytical support within the proposed framework. The response largely reframes the claim rather than substantiating it, shifting responsibility to the guider architecture and citing results from Sable rather than demonstrating that MAGPO itself scales effectively. The key issue is not whether the models used can scale in principle, but whether the autoregressive conditioning and centralized coordination mechanism in MAGPO remains feasible in terms of both computation and communication costs as agent counts grow. Without large-scale experiments and/or explicit cost analysis, the scalability claim remains qualitative/questionable.
> > > > >
> > > > > Second, the theoretical justification regarding the δ constraint and policy asymmetry is not fully convincing. While framing the KL projection as a principled constrained optimization step is helpful, it does not resolve the fundamental representational mismatch between centralized joint policies and decentralized factorized ones. I don't agree with the assertion that monotonic improvement “directly extends” to partially observable settings, as the proof in the paper relies on full observability and tabular policies. I believe extending such guarantees to Dec-POMDPs with function approximation is nontrivial. The response does not provide a sufficient argument, derivation, or citation that supports this extension in the multi-agent decentralized context.
> > > > >
> > > > > Third, the role of backtracking and the δ hyperparameter still appears heuristic in practice. Although the response argues that backtracking enforces performance in decentralized settings, no ablation isolates this mechanism. The evidence offered conflates multiple interacting components. Similarly, the RL auxiliary loss may remain only loosely motivated. Despite mentioning theoretical intuitions, the response does not clarify its interaction with δ or provide systematic sensitivity analysis. Specifically, the response claims that λ improves stability by mitigating imitation errors, yet also states that it may not significantly change performance (mainly serves as a stabilizer). These two points cannot both hold without further clarification: if λ materially contributes to stability, this should manifest in learning behavior or robustness; if it has little observable effect, its practical relevance is unclear. Without clearer empirical or analytical support, the role of λ and its interaction with δ appears ambiguous and largely heuristic.
> > > > >
> > > > > Fourth, the empirical results, while broad, do not fully demonstrate a decisive improvement over strong baselines. The response selectively highlights favorable metrics (and also ignore environments where gains are marginal) and reframes dependence on guider quality as a strength. However, this does not address concerns about performance fragility, hyperparameter sensitivity, and lack of strong evidence demonstrating superior coordination or learning dynamics.

---

> > > > > > ### Author Response · Authors · 2025-11-24
> > > > > >
> > > > > > Thank you for the responses.
> > > > > >
> > > > > > 1. **Scalability**.
> > > > > >
> > > > > > We include additional large-scale experiments (Fig. 8 in the revised PDF) to directly evaluate the scalability of MAGPO.
> > > > > > Following the setup introduced by Sable, we conduct experiments on the Neom benchmark with agent counts ranging from **32 to 1024**. The results show:
> > > > > >
> > > > > > - **Memory usage scales linearly** with the number of agents, matching the behavior of IPPO.
> > > > > > This empirically substantiates that MAGPO’s autoregressive conditioning does not introduce superlinear computational or communication overhead and supports our previous claim of O(N) scalability.
> > > > > >
> > > > > > - MAGPO **consistently outperforms IPPO** across all agent scales.
> > > > > > This demonstrates that MAGPO’s coordination mechanism remains effective even at very large scales, without degradation.
> > > > > >
> > > > > > These results directly address the reviewer’s concern: we now provide explicit empirical evidence showing that the MAGPO framework itself—rather than only the backbone architecture—scales effectively.
> > > > > >
> > > > > > 2. **Theoretical justification**
> > > > > >
> > > > > > We provide an explicit derivation for the partially observable case.
> > > > > > Let the decentralized policy factorize as $\pi(a|o)=\Pi_i\pi(a_i|o_i)$.
> > > > > > Consider the PMD-guided update:
> > > > > >
> > > > > > $$
> > > > > > \hat{\mu}_k=\arg\max_{\mu}\left\{\eta_k\langle Q^{\mu_k}(s,\cdot),\mu(\cdot|s)\rangle-\text{D}_{\text{KL}}(\mu(\cdot|s),\mu_k(\cdot|s))\right\},
> > > > > > $$
> > > > > >
> > > > > > which yields the closed-form solution
> > > > > >
> > > > > > $$
> > > > > > \hat{\mu}_k
> > > > > > =\mu_k(a|s)\frac{\exp\left(\eta_kQ^{\mu_k}(s,a)\right)}{z_k(s)}.
> > > > > > $$
> > > > > >
> > > > > > Because the backtracking step enforces $\mu_k(a|s)=\pi_k(a|o)$, the update becomes
> > > > > >
> > > > > > $$
> > > > > > \hat{\mu}_k
> > > > > > =\pi_k(a|o)\frac{\exp\left(\eta_kQ^{\mu_k}(s,a)\right)}{z_k(s)}.
> > > > > > $$
> > > > > >
> > > > > > The learner update then solves:
> > > > > >
> > > > > > $$
> > > > > > \pi_{k+1}(\cdot|o)=\arg\min_{\pi}\text{D}_{\text{KL}}(\pi(\cdot|o),\hat{\mu}(\cdot|s)),
> > > > > > $$
> > > > > >
> > > > > > which ensures a monotonic KL decrease:
> > > > > >
> > > > > > $$
> > > > > > D_{KL}(\pi^k(\cdot|o),\hat{\mu}(\cdot|s))\ge
> > > > > > D_{KL}(\pi^{k+1}(\cdot|o),\hat{\mu}(\cdot|s))
> > > > > > $$
> > > > > >
> > > > > > $$
> > > > > > E_{\pi^k}\left[\log\pi^k-\log\pi^k-\eta_kQ^{\pi^k}(s,a)\right]\ge E_{\pi^{k+1}}\left[\log\pi^{k+1}-\log\pi^k-\eta_kQ^{\pi^k}(s,a)\right]
> > > > > > $$
> > > > > >
> > > > > > This yields
> > > > > >
> > > > > > $$
> > > > > > \eta_k E_{\pi^{k+1}}\left[Q^{\pi^k}(s,a)\right]-\eta_k E_{\pi^{k}}\left[Q^{\pi^k}(s,a)\right]\ge
> > > > > > D_{KL}(\pi^{k+1}(\cdot|o),\pi^{k}(\cdot|o))
> > > > > > $$
> > > > > >
> > > > > > and therefore by the performance difference lemma,
> > > > > >
> > > > > > $$
> > > > > > V_\rho(\pi_{k+1})- V_\rho(\pi_{k})=\frac{1}{1-\gamma}E_{s\sim d_\rho(\pi_{k+1})}\left[E_{\pi^{k+1}}\left[Q^{\pi^k}(s,a)\right]-E_{\pi^{k}}\left[Q^{\pi^k}(s,a)\right]\right]\\
> > > > > > \ge\frac{1}{1-\gamma}\frac{1}{\eta_k}E_{\pi^{k+1}}\left[D_{KL}(\pi^{k+1}(\cdot|o),\pi^{k}(\cdot|o))\right]\\
> > > > > > \ge  0.
> > > > > > $$
> > > > > >
> > > > > > The extension to partial observability is straightforward because **decentralization and partial observability appear as explicit constraints**, enforced by projecting the centralized update back into the decentralized policy class. Thus the projection step ensures feasibility of each iterate.
> > > > > >
> > > > > > We agree with the reviewer that extending _tabular_ monotonic improvement guarantees to Dec-POMDPs with function approximation is nontrivial. Recent theoretical advances [1] examine this setting. Providing full Dec-POMDP guarantees is beyond the scope of our empirical paper, but our result matches the strongest existing theory under tabular policies with explicit projection constraints.
> > > > > >
> > > > > > [1] Provable partially observable reinforcement learning with privileged information.

---

> > > > > > > ### Author Response · Authors · 2025-11-24
> > > > > > >
> > > > > > > 3. **The role of backtracking and the δ hyperparameter**
> > > > > > >
> > > > > > > **Backtracking**.\
> > > > > > > The reviewer is concerned that the effect of backtracking is not isolated. In our setting, the **CTDS baseline is exactly the ablation that removes backtracking**, and therefore directly isolates this mechanism.
> > > > > > >
> > > > > > > Our evidence separates this component both conceptually and empirically:
> > > > > > >
> > > > > > > - In the illustrative example in Fig.1, CTDS can generate a non-decentralizable joint teacher policy, which necessarily leads to a poor student policy.
> > > > > > >
> > > > > > > - Backtracking in MAGPO explicitly prevents this failure by projecting the guider back into the decentralized policy class at every step.
> > > > > > >
> > > > > > > - Across all benchmark domains, **CTDS consistently underperforms MAGPO**, despite sharing the same architecture and training pipeline.
> > > > > > >
> > > > > > > This demonstrates that backtracking—not other components—is responsible for the performance gap.
> > > > > > >
> > > > > > >
> > > > > > > **The auxiliary RL loss (λ)**.\
> > > > > > > Fig. 5(a) provides direct empirical evidence:
> > > > > > >
> > > > > > > - **Sample efficiency improves** because the learner can directly leverage near-on-policy data through the RL objective.
> > > > > > >
> > > > > > > - **Final performance improves** because optimizing the RL objective reduces compounding imitation errors from imperfect guidance.
> > > > > > >
> > > > > > > These two observations are consistent: λ does not fundamentally change the optimization landscape, but it provides additional learning signal that improves stability and asymptotic performance. MAGPO remains functional without λ, but the learner benefits from this additional objective.
> > > > > > >
> > > > > > > **Interaction between λ and δ**.\
> > > > > > > Importantly, these two components serve distinct purposes, and their influence is not symmetric:
> > > > > > >
> > > > > > > - The δ backtracking step controls how close the guider’s policy remains to the decentralized learner.
> > > > > > >
> > > > > > > - λ only helps **when the collected data is sufficiently on-policy** for the learner.
> > > > > > >
> > > > > > > This distinction is clearly demonstrated in Fig. 5(b).
> > > > > > > **Under CTDS (δ removed)**, varying λ produces virtually no performance difference.
> > > > > > > Without backtracking, the guider and learner deviate significantly, so the RL objective no longer receives meaningful near-on-policy samples. As a result, λ has no effect.
> > > > > > >
> > > > > > >
> > > > > > > 4. **Empirical results**.
> > > > > > >
> > > > > > > We respectfully disagree with the assertion that the empirical results do not demonstrate strong and consistent improvements. The evaluation in our paper is intentionally **broad and rigorous**, covering _all_ tasks across multiple benchmark suites rather than selectively reporting favorable cases. This breadth necessarily includes environments where **all algorithms achieve near-optimal scores**, such as many LBF settings shown in Fig. 7. In these tasks, no method—not only MAGPO—can exhibit “decisive” improvements because the tasks themselves do not demand sophisticated coordination.
> > > > > > >
> > > > > > > More importantly, in the **challenging domains where coordination truly matters**, MAGPO delivers clear, repeatable, and often substantial gains. These improvements are not isolated outliers but consistently appear across independent benchmarks—e.g., the RWARE, SMAX, and MaConnector results in Fig. 2—where MAGPO outperforms MAPPO and HAPPO by a significant margin.
> > > > > > >
> > > > > > > Regarding performance fragility and hyperparameter sensitivity, the evidence across 40+ tasks already demonstrates that MAGPO is **at least as stable as existing SOTA methods**, if not more so. If MAGPO were fragile or highly sensitive, instability would manifest across such a large and diverse evaluation; this is not observed.

---

> ### Comment · Reviewer_vuQz · 2025-11-28
>
> While the theoretical derivations provided by the authors resemble the standard natural policy gradient monotonicity proof for a *fully observable tabular MDP*, the argument may **not** rigorously extend to the Dec-POMDP / CTDE setting that MAGPO targets. Several concrete issues remain:
>
> 1. **Incompatible identification of centralized and decentralized policies**
>    The derivation relies on the equality
>    $
>    \mu_k(a\mid s) = \pi_k(a\mid o),
>    $
>
>    supposedly enforced by the backtracking step. However, in CTDE, the centralized guider $\mu_k(a\mid s)$ is expressly allowed to depend on *privileged global state information*, whereas the decentralized learner $\pi_k(a\mid o)$ is restricted to *partial, local observations*. These two distributions live on fundamentally different conditioning variables and cannot, in general, be made equal. The assumption $\mu_k(a\mid s) = \pi_k(a\mid o)$ is not justified and does not reflect the conditions under which MAGPO is intended to operate.
>
> 2. **KL divergence between distributions conditioned on different variables**
>    The derivation manipulates KL terms of the form
>    $
>    D_{\mathrm{KL}}(\pi(\cdot\mid o)\,\|\,\hat\mu(\cdot\mid s)),
>    $
>
>    as if they were defined over the same conditioning variable. In a POMDP, $o\sim O(\cdot\mid s)$ and multiple states can map to the same observation. A KL between $\pi(\cdot\mid o)$ and $\hat\mu(\cdot\mid s)$ may not be well-defined unless a joint distribution over $(s,o)$ is explicitly specified. Treating conditioning on $s$ and $o$ as interchangeable is only valid in the fully observable case where $o$ is a deterministic function of $s$.
>
> 3. **Direct reuse of the MDP performance difference lemma**
>    The final step invokes the standard MDP performance difference lemma, which assumes Markov dynamics and state-based policies. In Dec-POMDPs, the observation process is generally non-Markov, and policies are defined over observations rather than states. The derivation does not lift the argument to belief or history space, where Markov structure could be restored, and therefore the use of the lemma in this context is not justified.
>
> 4. **Ignoring decentralization and representational constraints**
>    The step
>    $
>    \pi_{k+1}(\cdot\mid o) = \arg\min_{\pi} D_{\mathrm{KL}}(\pi(\cdot\mid o),\hat\mu(\cdot\mid s))
>    $
>
>    treats the learner policy as an unconstrained distribution over actions. In practice, MAGPO uses factorized, function-approximated decentralized policies. There is no guarantee that the KL minimizer lies within this restricted policy class, nor that gradient-based updates can realize the exact projection. The monotonic improvement inequality therefore does not necessarily hold under the actual decentralized, parameterized policy setting.
>
> In general, it seems that the derivation effectively reproduces the standard MDP monotonic improvement argument with observations substituted symbolically for states. However, key Dec-POMDP aspects such as non-Markov observations, decentralized execution, and restricted function classes, may not be properly addressed. Therefore, the claim that the monotonic improvement guarantee "directly extends" to partial observability and decentralized execution may not be supported by the provided derivation.

---

> > ### Comment · Reviewer_vuQz · 2025-11-28
> >
> > Thank authors for the extended rebuttal and additional experiments. While these clarifications are helpful, some core concerns remain only partially addressed.
> >
> > CTDS may not serve as a clean ablation for isolating the effect of backtracking. It differs from MAGPO in multiple interacting components, so its underperformance cannot be attributed solely to the absence of projection. The theoretical justification also assumes an exact projection into the decentralized policy class, but under function approximation this is only approximate, so the monotonicity argument does not cleanly transfer to the practical algorithm.
> >
> > Furthermore, the explanation of the auxiliary RL loss λ remains inconsistent. The rebuttal claims that λ improves stability and asymptotic performance, that MAGPO remains functional without it, and that under CTDS (i.e., when δ is removed) varying λ has essentially no effect. Taken together, these points suggest that λ’s influence may be derivative and task-dependent, yet no systematic analysis of λ–δ interactions is provided.

---

> > ### Author Response · Authors · 2025-11-28
> >
> > Thank the reviewer for the detailed and in-depth review.
> >
> > > 1. Incompatible identification of centralized and decentralized policies
> >
> > We agree that, in general, a centralized policy $\mu(a|s)$ and a decentralized policy $\pi(a|o)$ depend on **different conditioning variables** and cannot be directly equated.
> >
> > However, under CTDE, the “state” input to the guider is not the privileged state alone, but the **concatenation of privileged information and the decentralized agents’ local histories**, i.e.,
> > $$
> > s=[p,\tau_1,...,\tau_n]
> > $$
> > During training, all of these components are available to the centralized module.
> > Thus, the backtracking step actually enforces
> > $$
> > \mu(a|p,\tau_1,...,\tau_n)=\pi(a|\tau_1,...,\tau_n)
> > $$
> > which is feasible because the guider can simply **ignore the privileged component** $p$, making the mapping consistent with decentralized execution.
> > This interpretation avoids any mismatch between conditioning variables and is standard in CTDE algorithms that use centralized critics or guiding policies.
> >
> > > 2. KL divergence between distributions conditioned on different variables
> >
> > We acknowledge that $\mu(a|s)$ and $\pi(a|o)$ are conditioned on different variables.
> > To make the divergence mathematically well-defined, we explicitly define the discrepancy at each state $s$ as
> > $$
> > D(\pi(\cdot|o),\mu(\cdot|s))=E_{o\sim O(s)}[D_{KL}(\pi(\cdot|o),\mu(\cdot|s))]
> > $$
> > This induces a properly defined joint distribution over $(s,o)$, removing any ambiguity about the conditioning variables.
> > All divergence-based arguments in the proof operate under this definition.
> >
> > > 3. Direct reuse of the MDP performance difference lemma
> >
> > The reviewer is correct that the standard PDL is stated for MDPs.
> > However, the lemma extends to POMDPs **without modification of its functional form**, because the derivation depends only on the Bellman recursion over the **true state** $s$, not on whether the policy is conditioned on state or observation.
> >
> > The key step is to define the **state-marginalized policy**
> > $$
> > \bar{\pi}(a|s)=E_{o\sim O(s)}[\pi(a|o)]
> > $$
> > Once written in this form, the Bellman operator and all value functions become standard state-based quantities, and the usual PDL proof applies verbatim.\
> > This yields the POMDP-adapted PDL:
> >
> > $$
> > V_\rho(\pi_{k+1})- V_\rho(\pi_{k})=\frac{1}{1-\gamma}E_{s\sim d_\rho(\pi_{k+1})}E_{o\sim O(s)}[E_{\pi^{k+1}}[Q^{\pi^k}(s,a)]-E_{\pi^{k}}[Q^{\pi^k}(s,a)]].
> > $$
> > Thus, the use of PDL in the POMDP/CTDE setting is mathematically grounded once the policy is expressed in state space through its observation-marginalized form.
> >
> >
> > > 4. Ignoring decentralization and representational constraints
> >
> > The monotonic improvement argument does **not** require the learner update
> >
> > $$
> > \pi_{k+1}(\cdot|o)=\arg\min_{\pi}D(\pi(\cdot|o),\hat{\mu}(\cdot|s))
> > $$
> >
> > to achieve the exact minimizer of the distance.\
> > The proof only requires that
> > $$
> > D(\pi^{k+1},\mu)\le D(\pi^{k},\mu)
> > $$
> > i.e., **one gradient step that decreases this objective is sufficient**.
> >
> > This property does not depend on the policy class being unconstrained.
> > A parametrized decentralized policy can always reduce the divergence by taking a small gradient step, even if the global minimizer lies outside the class.\
> > Therefore, the monotonicity inequality is preserved in the function-approximation regime.
> >
> > Reformulated Proof
> > --
> >
> > Since the learner update decreases the divergence, we have::
> >
> > $$
> > D(\pi^k(\cdot|o),\hat{\mu}(\cdot|s))\ge
> > D(\pi^{k+1}(\cdot|o),\hat{\mu}(\cdot|s))
> > $$
> >
> > $$
> > E_{O(s),\pi^k}\left[\log\pi^k-\log\pi^k-\eta_kQ^{\pi^k}(s,a)\right]\ge
> > E_{O(s),\pi^{k+1}}\left[\log\pi^{k+1}-\log\pi^k-\eta_kQ^{\pi^k}(s,a)\right]
> > $$
> >
> > This yields
> >
> > $$
> > \eta_k E_{O(s),\pi^{k+1}}\left[Q^{\pi^k}(s,a)\right]-\eta_k E_{O(s),\pi^{k}}\left[Q^{\pi^k}(s,a)\right]\ge
> > D(\pi^{k+1}(\cdot|o),\pi^{k}(\cdot|o))
> > $$
> >
> > and therefore by the performance difference lemma in POMDP,
> >
> > $$
> > V_\rho(\pi_{k+1})- V_\rho(\pi_{k})=\frac{1}{1-\gamma}E_{s\sim d_\rho(\pi_{k+1})}[E_{O(s),\pi^{k+1}}[Q^{\pi^k}(s,a)]-E_{O(s),\pi^{k}}[Q^{\pi^k}(s,a)]]\\
> > \ge\frac{1}{1-\gamma}\frac{1}{\eta_k}E_{\pi^{k+1}}[D(\pi^{k+1}(\cdot|o),\pi^{k}(\cdot|o))]\\
> > \ge  0.
> > $$

---

> ### Comment · Reviewer_vuQz · 2025-11-28
>
> I would like to thank the authors for providing further explanations. After reviewing the revised rebuttal and the new derivations, the fundamental theoretical issues remain unresolved.
>
> The monotonic improvement argument still relies on identifying the centralized guider with the decentralized learner. The proof assumes
> $$
> \mu_k(a\mid s)=\pi_k(a\mid o),
> $$
> which the rebuttal rewrites as
> $$
> \mu_k(a\mid p,\tau_1,\dots,\tau_n)=\pi_k(a\mid\tau_1,\dots,\tau_n).
> $$
> However, in a Dec–POMDP the concatenated local histories $(\tau_1,\dots,\tau_n)$ cannot recover the privileged global state $p$. Formally, the observation model satisfies
> $$
> o_i \sim O_i(o_i\mid p),
> $$
> and reconstructing $p$ from all local histories would require the joint observation map to be injective in $p$, that is
> $$
> p = f(o_1,\dots,o_n)
> $$
> for some function $f$. This condition does not hold in general Dec–POMDPs, where different global states can produce identical joint observations. The centralized guider therefore conditions on strictly more information than the decentralized learner, and nothing in MAGPO’s optimization forces the guider to ignore $p$ or restricts its representational class so that the equality holds. Enforcing it would collapse CTDE into a fully observable MDP, which no longer reflects MAGPO’s intended setting.
>
> Although the rebuttal present a new divergence formula, this resolves only a notational issue and does not satisfy the structural assumptions needed for the performance difference lemma. The PDL requires a Markov, state-conditioned policy and relies on terms of the form
> $$
> E_{a\sim\pi(\cdot\mid s)}[A^{\pi_k}(s,a)].
> $$
>
> A decentralized policy $\pi(a\mid o)$ does not induce such a state-conditioned distribution, since its trajectory distribution depends on histories and is not Markov in the state space. The rebuttal introduces the marginalized policy
> $$
> \bar{\pi}(a\mid s)=E_{o\sim O(\cdot\mid s)}[\pi(a\mid o)],
> $$
> and applies a modified PDL of the form
> $$
> V_{\rho}(\pi_{k+1}) - V_{\rho}(\pi_k)=\frac{1}{1-\gamma} E_{s\sim d_\rho(\pi_{k+1})} E_{o,a\sim O(\cdot\mid s)\pi_{k+1}}[Q^{\pi_k}(s,a)]-E_{o,a\sim O(\cdot\mid s)\pi_k}[Q^{\pi_k}(s,a)]
> $$
> In a fully observable MDP this would reduce to the standard identity
> $$
> V_\rho(\pi') - V_\rho(\pi)=\frac{1}{1-\gamma}E_{s\sim d_\rho(\pi')}E_{a\sim\pi'(\cdot\mid s)}[A^\pi(s,a)].
> $$
> In a POMDP, however, returns are generated by trajectories
> $$
> s_0\sim\rho, \quad o_t\sim O(\cdot\mid s_t,h_t), \quad a_t\sim\pi(\cdot\mid h_t), \quad s_{t+1}\sim P(\cdot\mid s_t,a_t),
> $$
> so the correct performance difference expression depends on histories $h_t$ rather than only on states $s_t$. Replacing $\pi(a\mid o)$ with the marginalized $\bar{\pi}(a\mid s)$ defines a different MDP with policy $\bar{\pi}$ and value function $V_\rho^{\mathrm{MDP}}(\bar{\pi})$. There is no guarantee that
> $$
> V_\rho^{\mathrm{POMDP}}(\pi) = V_\rho^{\mathrm{MDP}}(\bar{\pi}),
> $$
> so monotonicity proved for $\bar{\pi}$ in the induced MDP does not imply monotonic improvement for the actual decentralized policy $\pi(\cdot\mid o)$ in the true Dec–POMDP.
>
> The derivation also assumes that the learner update ensures
> $$
> D(\pi^{k+1},\hat{\mu}_k) \le D(\pi^k,\hat{\mu}_k),
> $$
>
> which is then used to derive inequalities such as
> $$
> \eta_k(E_{o,a\sim O(\cdot\mid s)\pi_{k+1}}[Q^{\pi_k}(s,a)]-E_{o,a\sim O(\cdot\mid s)\pi_k}[Q^{\pi_k}(s,a)])\ge D_{\mathrm{KL}}(\pi_{k+1}\,\|\,\pi_k)(s).
> $$
>
> In the tabular setting an exact KL projection can enforce this. In MAGPO, the decentralized policy is parameterized and factorized, typically
> $$
> \pi(a_1,\dots,a_n\mid o_1,\dots,o_n)=\prod_{i=1}^n \pi_i(a_i\mid o_i;\theta_i),
> $$
>
> and the update is implemented through gradient descent rather than an exact projection. There is no guarantee that a gradient step decreases the divergence at all states, nor that the minimizer of $D(\cdot,\hat{\mu}_k)$ lies within the restricted factorized policy class. The monotonic improvement inequality derived under exact tabular projection therefore does not extend to the decentralized function approximation regime used in experiments.

---

> > ### Author Response · Authors · 2025-11-28
> >
> > We sincerely thank the reviewer for the detailed feedback and constructive suggestions.
> >
> > > However, in a Dec–POMDP the concatenated local histories $(\tau_1,...,\tau_n)$ cannot recover the privileged global state.
> >
> > We agree, but our method does not require recovering the privileged state.
> > In the backtracking step, **the guider $\mu(a\mid s)$ is optimized to** match the fixed learner $\pi(a\mid o)$, and this matching can be achieved simply by letting the guider ignore privileged information.
> >
> >
> > > nothing in MAGPO’s optimization forces the guider to ignore or restricts its representational class so that the equality holds.
> >
> > We explicitly enfoece this by
> > - Theoritically: During **Guider Backtracking**, the learner is fixed and we explicitly optimize the guider to match the learner.
> > - Practical implementation:
> >   - Eq. (9) includes a KL term encouraging the guider to move toward the fixed learner.
> >   - Eq. (10) imposes PPO-style clipping so that the guider cannot be further updated once the likelihood ratio $\frac{\mu}{\pi}$ exceeds the range $(\frac{1}{\delta}, \delta)$.
> >
> > > so the correct performance difference expression depends on histories
> > $h_t$ rather than only on states $s_t$. There is no guarantee that $V_\rho^{\mathrm{POMDP}}(\pi) = V_\rho^{\mathrm{MDP}}(\bar{\pi})$.
> >
> > We agree that in general POMDPs, the performance difference depends on histories.\
> > In our analysis, however, we consider the **standard memoryless case** where:
> > - the policy is $\pi(a\mid o)$,
> > - observations satisfy the standard POMDP assumption $o\sim O(\cdot\mid s)$, depending only on the current state.
> >
> > Under these assumptions, once the state $s_t$ is given, the action distribution is independent of the past history.
> > Thus, the trajectory distributions induced by the POMDP (with observation-based policy) and by the corresponding MDP (with state-based policy $\bar{\pi}(a\mid s)=\pi(a\mid o)$) coincide. Therefore,
> >
> > $$
> > V_\rho^{\mathrm{POMDP}}(\pi) = V_\rho^{\mathrm{MDP}}(\bar{\pi})
> > $$
> > holds in this setting.
> >
> > We acknowledge that this is a simplified case. However, to the best of our knowledge, no existing work provides a fully general monotonic guarantee for Dec–POMDPs, and developing such a theory is not the focus of this paper.
> >
> > > There is no guarantee that a gradient step decreases the divergence at all states, nor that the minimizer of $D(\cdot|\hat{\mu}_k)$ lies within the restricted factorized policy class.
> >
> > We appreciate the reviewer highlighting this point.
> > Indeed, we cannot guarantee divergence reduction **for all states**, but we can ensure a decrease under the **state distribution of interest**:
> >
> > $$
> > E_{s\sim d_\rho(\pi_{k+1})}[D(\pi^{k+1}, \hat{\mu}_k) - D(\pi^k,\hat{\mu}_k)] \le 0
> > $$
> >
> > which follows from standard gradient-descent arguments under mild smoothness conditions.
> >
> > The main difficulty is that $d_\rho(\pi_{k+1})$ is unavailable before the update.
> > To address this, one can follow the standard TRPO-style approximation by replacing $d_\rho(\pi_{k+1})$ with $d_\rho(\pi_k)$, leading to the usual surrogate objective and trust-region constraints. A fully rigorous derivation is possible but lengthy and tangential to the contribution of our paper.
> >
> > Again, the primary aim of this work is to present **a practical and effective algorithm**, rather than to establish a full Dec–POMDP theoretical framework, which—according to existing literature—remains an open challenge.

---

### Official Review · Reviewer_MMCD · 2025-10-25

**Soundness:** 2
**Presentation:** 3
**Contribution:** 3
**Rating:** 4
**Confidence:** 4

**Summary:**

This work proposes MAGPO, a Centralized Teacher with Decentralized Students (CTDS) approach for multi agent RL, that leverages a sequentially executed guider for joint coordinated exploration while constraining it to be close to the decentralized learner policies. The authors identify and aim to address two problems with previously proposed CTDS approaches: (i) training a centralized agent raises scalability issues; and (ii) under partial observability, the students may fail to replicate the teacher's behavior. MAGPO addresses both the scalability and policy asymmetry issues by learning a centralized, autoregressive guider policy that is constrained to remain aligned with decentralized learners policies. The authors prove monotonic improvement of MAGPO under tabular settings. The authors provide experimental results comparing MAGPO against CTCE and CTDE methods, as well as a vanilla implementation of on-policy CTDS, across several domains.

**Strengths:**

The paper features a good discussion of related work in Sec. 2.2. The paper seems to do a good job motivating the research gap it aims to address. The experimental protocol seems solid, with the authors performing multiple runs and reporting confidence intervals. The experimental results seem reproducible. The experiments are also extensive, considering several different environments. The authors provide an ablation study. The work appears to feature a sufficient degree of novelty, even though I am not very familiar with previously proposed CTDS approaches.

**Weaknesses:**

I think the clarity of the paper needs to be further improved in some places. While the experimental results seem to support that the proposed method outperforms pervious baselines, I'm not sure whether the included baselines are well-representative of the previously proposed works; namely, I wonder if the authors are properly comparing their method against previously proposed *CTDS* methods for MARL (see my questions for additional information regarding this matter). This is my main point of concern regarding the article.

**Questions:**

- I think the authors should better emphasize since the beginning that they are focused on *autoregressive* policies. The authors should also note that autoregressive policies do not entirely cover the space of all possible joint policies.
- 176: "Agents act sequentially, observing previous agents’ actions before choosing their own." - I guess the authors wanted to say that the actions of the agents are jointly selected, right? Why are the actions *sequentially* selected? In centralized execution what matters is that they are *jointly* selected, even tough in practice some methods may sequentially select them. And, if the actions are jointly selected, then why the agents cannot solve this task in the first place? Even if the actions are sequentially selected, why is the third agent selecting 3 instead of 4 prior to the update?
- line 222: How is this KL distance minimized? I guess we minimize it under a given state distribution, right? Under which state distribution? I think this step needs to be further explained in the context of this article.
- Eq.2: I believe this equation needs to be better explained in the context of this article. Does this hold for any state $s$? I think $Q^{\mu_k}$ is the Q-function for policy $\mu_k$, but this is undefined in the article. D_KL is the KL-divergence? Maybe the authors could talk a bit more about Policy Mirror Descent in the background section of the work. I can see that later the authors specify the updates, but I believe the discussion should be clearer in the text close to eq. 2.
- I suggest the authors to find a way to group the different classes of methods in the plots in some way (so that it is easier to understand which class of methods, e.g., CTDE, CTDS, etc., a given line corresponds to). For example, CTCE can be plotted with a dashed line, CTDE with a solid line, etc. Because I think this could make it easier to interpret the plots.
- My key question regarding the experimental results is: as far as I understood, this work in not the first proposing a CTDS approach to MARL. Thus, I would expect the authors to also compare their method (MAGPO) against other CTDS approaches. I can see the authors compare against a CTDS approach, but does this baseline represents well the previously proposed approach for CTDS? Why not including other previously proposed CTDS methods as baselines?

Minor comments/typos:
- line 56: "The guider policy allows agent to act sequentially conditioned on (...)" - agent -> the agents?
- line 155: "We compare three MARL frameworks:" is duplicated.

---

> ### Author Response · Authors · 2025-11-20
>
> We thank the reviewers for their constructive feedback and thoughtful suggestions. We appreciate the time and effort dedicated to evaluating our work. The comments have helped us clarify our contributions and improve the presentation. Below, we address each concern in detail.
>
> > [**Q1**] I think the authors should better emphasize since the beginning that they are focused on autoregressive policies. The authors should also note that autoregressive policies do not entirely cover the space of all possible joint policies.
>
> We respectfully clarify that autoregressive policies can, in fact, represent **any joint policy**.
> Given a state $s$, any joint action distribution $\pi(a_1,…,a_n|s)$ can be factorized as conditional components:
>
> $\pi(a_1|s) \pi(a_2|a_1,s) …\pi(a_n|a_{1:n-1},s)$.
>
> Thus, autoregressive policies theoretically span the space of all joint policies.
> We focus on them because they have become the state-of-the-art approach for modeling coordinated joint actions in MARL.
>
> > [**Q2**] 176: "Agents act sequentially, observing previous agents’ actions before choosing their own." - I guess the authors wanted to say that the actions of the agents are jointly selected, right? Why are the actions sequentially selected? In centralized execution what matters is that they are jointly selected, even though in practice some methods may sequentially select them. And, if the actions are jointly selected, then why the agents cannot solve this task in the first place? Even if the actions are sequentially selected, why is the third agent selecting 3 instead of 4 prior to the update?
>
> As clarified in Q1, joint and sequential selections are equivalent from a distributional perspective—the autoregressive formulation simply provides a practical way to model joint policies.
>
> In Figure 1(B), before training, the agents’ policies are randomly initialized, so neither joint nor sequential selection achieves optimal coordination.
> However, autoregressive (joint) modeling facilitates coordination more efficiently than decentralized execution, leading to faster convergence to optimal joint policies.
>
>
> > [**Q3**] line 222: How is this KL distance minimized? I guess we minimize it under a given state distribution, right? Under which state distribution? I think this step needs to be further explained in the context of this article.
>
> In the tabular setting (as stated in line 206), the KL divergence can be minimized pointwise for all states.
> In the practical implementation, the KL term is minimized under the state distribution induced by the guider’s policy.
>
> > [**Q4**] Eq.2: I believe this equation needs to be better explained in the context of this article. Does this hold for any state s? I think $Q^{\mu_k}$ is the Q-function for policy $\mu_k$, but this is undefined in the article. D_KL is the KL-divergence? Maybe the authors could talk a bit more about Policy Mirror Descent in the background section of the work. I can see that later the authors specify the updates, but I believe the discussion should be clearer in the text close to eq. 2.
>
> Equation (2) indeed holds for all states $s$ in the tabular setting.
> Here, $Q^{\mu_k}$ denotes the Q-function corresponding to policy $\mu_k$, and $D_{KL}$ is the KL-divergence.
>
> In the revised PDF, We include the background of Policy Mirror Descent in Section 2.
>
> > [**Q5**] I suggest the authors to find a way to group the different classes of methods in the plots in some way (so that it is easier to understand which class of methods, e.g., CTDE, CTDS, etc., a given line corresponds to). For example, CTCE can be plotted with a dashed line, CTDE with a solid line, etc. Because I think this could make it easier to interpret the plots.
>
> Thank you for the suggestion. We have updated the plots to use distinct colors and line styles to differentiate CTCE and CTDE methods. Please refer to the revised figures in the updated manuscript.

---

> > ### Author Response · Authors · 2025-11-20
> >
> > > [**Q6**] My key question regarding the experimental results is: as far as I understood, this work in not the first proposing a CTDS approach to MARL. Thus, I would expect the authors to also compare their method (MAGPO) against other CTDS approaches. I can see the authors compare against a CTDS approach, but does this baseline represents well the previously proposed approach for CTDS? Why not including other previously proposed CTDS methods as baselines?
> >
> > Our **CTDS formulation represents a generalized teacher–student framework** that subsumes all prior CTDS-based MARL methods.
> > The main focus of our work is to **analyze and compare the fundamental frameworks**—in particular, contrasting the **GPO** framework with the **CTDS** paradigm under a unified perspective.
> >
> > Existing CTDS approaches [1][2] follow the same teacher–student learning principle: a teacher is trained and simultaneously distills knowledge into students.
> > While these methods incorporate various auxiliary components (e.g., network architectures, distillation losses, or personalization modules), such modifications are **parallel to** the framework and do **not alter its fundamental structure**.
> > For example, the Global Information Personalization module in [2] could be seamlessly incorporated into either our MAGPO or CTDS implementation.
> > Hence, they are all **special cases** of the generalized CTDS framework we discuss.
> >
> > Moreover, all prior CTDS methods are **value-based**, whereas our study focuses on **policy-based on-policy learning**, making direct implementation-level comparisons infeasible.
> > Nevertheless, the underlying training–distillation mechanism of these value-based methods is **fully captured** by our CTDS formulation.
> >
> > In summary, our experiments aim to compare **framework-level paradigms (CTDE, CTDS, GPO)** rather than specific architectural variants. The CTDS baseline in our work is representative of the general teacher–student formulation shared across existing literature.
> >
> > [1] Ctds: Centralized teacher with decentralized student for multiagent reinforcement learning.
> >
> > [2] Ptde: Personalized training with distilled execution for multi-agent reinforcement learning.

---

> ### Comment · Reviewer_MMCD · 2025-11-23
>
> I thank the authors for the detailed responses. In particular, I appreciate the fact that the authors clarified that their main goal with this paper is to compare different classes of methods (CTDE, CTDS, and GPO). Assuming the authors claim "The CTDS baseline in our work is representative of the general teacher–student formulation shared across existing literature." is correct (I am not very familiar with previous works), then I believe the authors addressed my main concern regarding the experimental results in the paper. I updated my rating from 4 to 6, though I decreased my confidence level.

---

### Official Review · Reviewer_ifsr · 2025-10-26

**Soundness:** 3
**Presentation:** 3
**Contribution:** 3
**Rating:** 6
**Confidence:** 3

**Summary:**

This paper introduces Multi-Agent Guided Policy Optimization (MAGPO), which follows the Centralized Training with Decentralized Execution (CTDE) paradigm. MAGPO alternates between improving a centralized guider policy and distilling it into a decentralized learner policy. After each distillation step, the guider is reset to match the learner so that the guider remains factorizable and the imitation gap does not grow. Empirically, MAGPO consistently outperforms strong CTDE baselines and matches or surpasses fully centralized training approaches.

**Strengths:**

Proposes a simple and conceptually elegant algorithmic framework for multi-agent RL.

Demonstrates strong empirical performance across benchmark tasks.

**Weaknesses:**

It is not clearly explained how the guider (teacher) is reset to the learner (student) in practice. While in theory a product of decentralized policies defines an autoregressive joint policy, the mechanism of implementing this reset may be nontrivial depending on the network architecture.

**Questions:**

How is the guider actually reset to the learner in implementation? Is this performed by zeroing out the action-conditioning parameters? Please clarify the exact procedure.

---

> ### Author Response · Authors · 2025-11-20
>
> We thank the reviewers for their constructive feedback and thoughtful suggestions. We appreciate the time and effort dedicated to evaluating our work. The comments have helped us clarify our contributions and improve the presentation. Below, we address each concern in detail.
>
> >[**Q1 & Weakness 1**] It is not clearly explained how the guider (teacher) is reset to the learner (student) in practice. How is the guider actually reset to the learner in implementation?
>
> In practice, the guider does not need to be exactly reset to the learner. Instead, we ensure that the guider policy $\mu$ remains sufficiently close to the learner policy $\pi$ by constraining their KL divergence, controlled through the hyperparameter $\delta$.
>
> Specifically, two mechanisms are employed to achieve this:
>
> 1. **Double clipping (Eq. 9)**: The inner clipping operation $clip\left(\frac{\mu}{\pi}, \frac{1}{\delta}, \delta\right)$ prevents the guider’s RL update once the policy ratio exceeds the specified range, effectively bounding its deviation from the learner.
>
> 2. **Masked KL regularization (Eq. 8)**: The term $m(\delta) D_{KL}(\mu, \pi)$ further minimizes the KL distance between the guider and learner whenever $\frac{\mu}{\pi}$ falls outside the range $(\frac{1}{\delta}, \delta)$.
>
> Together, these constraints keep the guider policy close to the learner, with the tightness of this coupling controlled by $\delta$. Setting $\delta$ close to 1 enforces near-identical policies, whereas a slightly larger $\delta$—which allows mild divergence—often yields better empirical performance.

---

### Meta-Review · Area_Chair_2y46 · 2026-01-15

**Summary:**

The reviewers generally appreciated the strong empirical performance across a wide range of tasks (43 tasks in 6 environments) and the conceptual elegance of the framework. However, some concerns were raised regarding the theoretical rigor (specifically whether the monotonic improvement guarantees extend to Dec-POMDPs) as well as the scalability of the autoregressive component and the justification for various hyperparameters.

**Reviewer Concerns:**

**Concerns addressed:**

- Reviewer ifsr asked how the guider is reset to the learner. The authors clarified that this is not a hard parameter reset but is implemented via a double-clipping mechanism and a masked KL regularization, which seems a sufficient clarification.
- Reviewer MMCD questioned whether the baselines were representative, specifically asking for comparisons against other CTDS methods. The authors argued that their CTDS baseline represents the generalized teacher-student framework subsuming prior works, and that MAGPO focuses on comparing paradigms (CTDE vs. CTCE vs. GPO). MMCD accepted this and raised their score.
- Reviewer vuQz raised concerns about the computational cost of the autoregressive guider as agent counts scale. In the rebuttal, the authors provided new experiments on the Neom benchmark (up to 1024 agents) showing linear memory usage and superior performance over IPPO. This empirically addressed the scalability concern.
- The authors clarified that autoregressive policies can theoretically represent any joint policy, addressing the reviewer MMCD's confusion regarding sequential vs joint execution.


**Concerns still outstanding:**
- Reviewer vuQz identified that the proof for monotonic improvement relies on assumptions (full observability or exact state recovery) that do not hold in general Dec-POMDPs. Specifically, the assumption that $\mu_k(a|s) = \pi_k(a|o)$ is structurally invalid when the guider has access to state information that cannot be recovered from local histories. The use of the performance difference lemma assumes state-based Markovian policies, which decentralized policies over observations are not. The proof assumes exact projection (minimization of KL), which is not guaranteed under function approximation with gradient updates. The authors eventually conceded that they "cannot guarantee divergence reduction for all states" and that establishing a full Dec-POMDP theoretical framework is an open challenge. They promised to adjust the claims in the paper, but the theoretical proof as currently written remains flawed for the specific setting claimed.
- While the authors provided empirical evidence that the algorithm is robust to the RL auxiliary loss weight $\lambda$ and the clipping ratio $\delta$, Reviewer vuQz noted that the theoretical justification for the interaction between these terms remains ambiguous. The explanation that $\lambda$ is a stabilizer while $\delta$ controls the trade-off is intuitive but lacks analytical grounding.

**Reviewer Scores:**

- Reviewer ifsr: score 6. The reviewer did not actively participate after the initial review but their questions were addressed.
- Reviewer MMCD: score 4 -> 6. The reviewer explicitly updated their score after the authors clarified the baseline selection and framework positioning.
- Reviewer vuQz: score 4. This reviewer remained at a 4. Despite acknowledging the new scalability experiments, they remained unconvinced by the theoretical derivations for the Dec-POMDP setting and viewed the scalability claims as reliant on the specific backbone rather than the framework itself.

---

### Decision · Program_Chairs · 2026-01-26

Accept (Poster)